



# Towards identification of critical rainfall thresholds for urban pluvial flooding prediction based on crowdsourced flood observations

**Christian Bouwens[1], Marie-Claire ten Veldhuis[1], Marc Schleiss[1], Xin Tian[1], Jerôme Schepers[2]**

11-1-2018

[1] Faculty of Civil Engineering and Geosciences, Delft University of Technology, Stevinweg 1, 2628 CN Delft, Netherlands
[2] Water Department, City Management, Municipality of Rotterdam, Wilhelminakade 179, 3072AP Rotterdam, Netherlands

*Correspondence to: Christian Bouwens (C.J.L.Bouwens@student.tudelft.nl) and Xin Tian (X.Tian@tudelft.nl)*

## Abstract

Urban drainage systems are challenged by both urbanization and climate change, intensifying urban pluvial flooding impacts. Urban pluvial flooding impacts can be reduced by improving infrastructure and operational urban flood management strategies. This study investigated the relation between heavy rainfall and urban pluvial flooding in Rotterdam with the aim to identify parameters and thresholds that can be used as predictors of urban pluvial flooding. The focus of the investigation was on an area of 16 km². Datasets for this research included historical crowdsourced flooding reports, overflow pumping volumes, open spatial data and rainfall data at different temporal and spatial resolutions. Threshold values, (which can be used as part of early warning systems,) were derived from historical flooding data and rainfall depths over sub daily durations. Threshold values of rainfall depth were found to be 6 mm (±3 mm) in 15 min and 11 mm (±6 mm) in 60 min. Surprisingly, the derived thresholds are only approximately half of the theoretical drainage system design capacity. Furthermore, a threshold value of 70% (± 4%) imperviousness was found above which flooding incidents significantly increase. Results also suggested a strong dependence on spatial aggregation scale, as it highly influences correlation coefficients and parameter density values. Elevation differences did not appear to contribute to urban pluvial flooding, based on a flow path analysis for the study area. Finally, we showed that antecedent rainfall does not explain additional variance in reports, meaning it is not an influential urban pluvial flooding predictor, at least not on a daily temporal resolution.

*Keywords: Urban pluvial flooding, rainfall variability, early warning thresholds, crowdsourced, open spatial data*

## 1. Introduction

### 1.1 Urban pluvial flooding and flood management

Urban pluvial flooding is caused by both natural and human factors (Tingsanchali, 2012), e.g. heavy rainfall in combination with high building densities and impervious surfaces. Urbanization and climate change intensify urban pluvial flooding impacts, as a combination of intensified rainfall extremes and expanded impervious surfaces result in larger storm water runoff volumes (Gaitan et al., 2016; Semadeni-Davies et al., 2008; ten Veldhuis et al., 2011; Yao et al., 2016). Impact on the urban environment, including decreased runoff time and larger storm water volumes, lead to larger peak flow volumes and thereby greater flooding risks (Butler and Davies, 2004; Griffiths, 2017). Urban pluvial flooding seriously threatens infrastructure, people and the environment, which can lead to economic losses (Hurford et al., 2012; Spekkers et al., 2015). Urban pluvial flooding impacts (e.g. flooded streets/tunnels, sewer overflow) can be reduced by improving operational urban flood management strategies (ten Veldhuis et al., 2011; Wheater and Evans, 2009). Urban flood management is expressed as optimization and maintenance of the overall functioning of a city through utilizing opportunities like cost-effective multi-functional solutions (Zevenbergen et al., 2017).





## 1.2 Approaches to quantifying urban pluvial flooding occurrence and impacts

Urban pluvial flooding can be investigated by analyzing historical flooding data, open spatial data, antecedent moisture and hydrodynamic modelling.

A widely used approach is to analyze urban pluvial flooding through hydrodynamic models (Van Bijnen et al., 2012; Candela and Aronica, 2016; Guerreiro et al., 2017; Löwe et al., 2017; Ochoa-Rodriguez et al., 2015; Palla et al., 2016; Thorndahl et al., 2016). Hydrodynamic models include the interaction between overland flows and sewer flows and are usually based on the Saint Venant equations to model flood wave propagation (Ochoa-Rodriguez et al., 2015; Palla et al., 2016; Thorndahl et al., 2016). However, the applicability of hydrodynamic models are hindered by many physical and computational limitations. For instance, uncertainties arise since in-sewer defects, like sedimentation, corrosion and root intrusion in pipes, are not captured in hydrodynamic models. Not taking these factors into account causes over-estimation of sewer system performance and under-estimation of urban flood risks in modelling urban pluvial flooding (Van Bijnen et al., 2012; ten Veldhuis et al., 2011).

A different approach is to analyze urban pluvial flooding using historical flooding record data. Flooding record data usually originates from either crowdsourced reports, such as complaint calls or questionnaires, or maintenance reports, e.g. gully pot or sewer inspections. Report data allows system operators to identify dominant failure mechanisms, flood-prone areas or urban flood risks (Arthur et al., 2008; Caradot et al., 2011; Cherqui et al., 2015; Gaitan et al., 2015, 2016; Spekkers et al., 2015; ten Veldhuis et al., 2011; Verstraeten and Poesen, 1999). Therefore, this approach is well suited to capture small-scale in-sewer defects. An example is given by insurance claims, originating from rainfall related damage in Rotterdam, used as a measure of urban pluvial flooding. This has resulted in the identification of important failure mechanisms and a rainfall threshold value around 10 mm/60min, above which damage to private property by street flooding increases significantly (Spekkers et al., 2015). Also, sewer system maintenance records have been used to identify flood-prone locations and dominant failure mechanisms in the French city of Bordeaux (Cherqui et al., 2015). A final example is given by Municipal call center data (crowdsourced flooding reports), which have been used to determine dominant failure mechanisms and to quantify urban flood risk probabilities in the cities of Prinsenbeek and Haarlem (ten Veldhuis et al., 2011).

According to many scientific investigations, open spatial data can also be used to analyze urban pluvial flooding. This data is actually also suitable for hydrodynamic modelling. For example, all hydrodynamic models rely on elevation data to characterize overland flow (Candela and Aronica, 2016; Van Dijk et al., 2014; Guerreiro et al., 2017; Palla et al., 2016; Thorndahl et al., 2016). Digital Elevation Models (DEM) can be used to delineate urban (sub)watersheds with GIS software to analyze preferential overland flow paths of surface runoff. This allows us to accurately investigate which parameters influence the propagation of surface runoff and the origin of flood prone areas (Diaz-Nieto et al., 2012; Gaitan et al., 2015, 2016). A flow path analysis for a district in Rotterdam has shown that flooding reports do not have a preferential location downstream of overland flow paths (Gaitan et al., 2015). In a more recent study, it is found that imperviousness and distance to watershed outflow point can be used to explain up to 50% of the spatial variability of flooding reports, while population density appears to be a weaker predictor of flooding incidents (Gaitan et al., 2016).

Another potential urban pluvial flooding predictor is antecedent moisture originating from preceding rainfall events. It expresses catchment wetness and is usually approximated through antecedent soil moisture or rainfall (Brocca et al., 2009). Antecedent moisture influences physical processes like infiltration, wetting of surfaces and depression storage, for both paved and unpaved surfaces. These processes lead to reduced or delayed peak volumes of runoff, assuming an initially dry urban environment (Butler and Davies, 2004; Griffiths, 2017), hence are expected to influence urban pluvial flooding. However, not much scientific consensus exists on this topic. Most studies focus on the process of infiltration capacity (or, in other words, antecedent soil moisture). Antecedent soil moisture in urban environments is commonly assumed not to have an effect on the hydrologic response of an urban catchment (Hurford et al., 2012; Wheater and Evans, 2009). Some investigations show weak correlations between antecedent soil moisture and storm water runoff in urban environments (Kjeldsen et al., 2013; Pitt et al., 2008; Yang et al., 2016). Another investigation focuses on rainfall characteristics and storm event hydrologic response for 7 urbanized catchments, where no strong correlation is found with antecedent moisture (Smith et al., 2013). However, in some investigations a significant correlation between antecedent soil moisture and runoff production is found (Shi et al., 2007; Shuster et al., 2008). This is on one hand based on a laboratory experiment (Shuster et al., 2008) and on the other hand on an investigation of the Buji River Basin in China (Shi et al., 2007). Both studies find increased amounts of runoff with wetter antecedent soil moisture conditions.

Obviously, rainfall information is essential to quantify urban pluvial flooding. Heavy rainfall events, which are most relevant for flooding, are rare, limiting the type of observation-based analyses that can be performed.



Numerous scientific investigations emphasize the importance of variable temporal resolutions when the urban

| Dataset | Source | Description | Spatial/temporal resolution | Original unit |
|---|---|---|---|---|

hydrologic response is relevant (Blenkinsop et al., 2017; Ochoa-Rodriguez et al., 2015; Romero et al., 2011; Simões et al., 2015). In an urban environment, the critical response time is in the range of minutes to hours (Romero et al., 2011). Therefore, variable sub daily temporal resolutions are important to enhance our
understanding of climate extremes (Blenkinsop et al., 2017). Research has shown that these variable resolutions influence hydrodynamic model outputs (Ochoa-Rodriguez et al., 2015). A variable spatial resolution of rainfall also appears to be a significant parameter in the determination of flood probabilities (Simões et al., 2015). Recently, an alternative framework is proposed, in which the variable of interest is the time needed to accumulate a fixed amount of rainfall or flow instead of a variable amount over a fixed arbitrary time (Ten Veldhuis and Schleiss,
2017). From this, it follows that rainfall and hydrological response are strongly scale dependent, indicating the relevance of analysis on varying temporal- and spatial resolution.

### 1.3 Objective of this study

The aim of this study is to investigate heavy rainfall and its impact on urban pluvial flooding for the city of Rotterdam. This is addressed through the analysis of historical flooding data, (open) spatial data and rainfall data.
Crowdsourced flooding reports (hereafter referred to as 'reports'), as well as overflow pumping data (hereafter referred to as 'OP'), are considered as indicators of urban pluvial flooding. OP is actually an indicator of system overloading, but it is an essential measure to prevent urban pluvial flooding and therefore it can be used as an indicator of (potential) flooding, as will be explained in Sect. 2.2. Combining these data types with rainfall data allows us to derive thresholds that can be used as part of early warning systems. As touched upon in Sect. 1.1 and
1.2, there are many parameters affecting the relation between rainfall and urban pluvial flooding. Therefore, an important aspect of this investigation is to determine which parameters can be used as urban pluvial flooding predictors.

The focus of the study is on the city center area of Rotterdam, characterized by a high degree of imperviousness and high flooding incidence, as indicated by previous research (Bouwens, 2017).

The following three research questions are addressed:

1. What is the relationship between historical flooding observations and rainfall?
2. How do imperviousness and elevation differences influence urban pluvial flooding?
3. What is the influence of antecedent rainfall on urban pluvial flooding?

The outline of this study is as follows. In Sect. 2, the available datasets are explained and the study area is described.
Sect. 3 addresses analysis methods and assumptions, which are used to elaborate the research questions. Sect. 4 shows the results of the investigation, which are subsequently discussed in Sect. 5. Finally, Sect. 6 provides the conclusions          of             the             investigation.

## 2. Data

### 2.1 Datasets

An overview of the datasets used in this study is provided in Table 1. To address the influence of imperviousness on urban pluvial flooding, land use data (PDOK, 2016) were obtained. Elevation data (1x1 m resolution, ±10 cm vertical accuracy) were used from the city of Rotterdam. Also, population statistics (Dutch Central Bureau of Statistics (CBS, 2015)) and open street maps (Publieke Dienstverlening op de Kaart (PDOK, 2016)) were obtained. Historical flooding data were selected for Rotterdam in the form of crowdsourced flood reports. Reports contain
information about date, flooding category, problem description and GPS coordinates. A total of 8932 water related reports for the city center area of Rotterdam for the period 2010-2016 were obtained. Additionally, OP data were selected, which consist of overflow volumes per sewer district per day. OP is considered a measure of potential urban pluvial flooding, as it is performed to avoid CSO (more explanation in Sect. 2.2). Three sewer districts, that have the ability to perform OP, overlap with the study area. These sewer districts (Figure 1b), which are called
'West', 'Central' and 'East', were included in the analysis. Since historical flooding data was available only at daily resolution, every event in this study was defined within a 24 hour window. National Rainfall Radar images (5 minute rainfall depth measurements, 1x1 km resolution), corrected with ground station measurements and compiled from radar images from the Dutch KNMI, German DWD and Belgium KMI, were used as rainfall data (Nelen & Schuurmans, 2017b).



| Flooding reports | City of Rotterdam | Tabulated and geo-coded crowdsourced flooding reports | Point shape file with date stamp | - |
|---|---|---|---|---|
| Overflow pumping | City of Rotterdam | OP, overflow pumping volumes | Available per sewer district | m³/day |
| Rainfall | National Rainfall Radar | Rainfall raster images, corrected based on ground stations, KNMI, DWD and KMI data | 1 km² cells, 5 min intervals | mm |
| Land use | PDOK | Characterized surface areas in pre-defined Land use classes from Basisregistratie Grootschalige Topografie (BGT) | Polygons | - |
| DEM | City of Rotterdam | Elevation map (no water bodies or buildings) | 1 m² cells | m |
| Population | CBS | Population statistics | Available per sub district | - |
| Open street maps | PDOK | Background maps for QGIS analyses | 0.25 m² cells | - |

**Table 1. List of datasets used for analysis, including description, resolution and unit.**

## 2.2 Study area description

The analysis mainly focused on the administrative districts of Rotterdam Centre, Noord and Delfshaven. To compare with these heavily urbanized areas, we add the district of Kralingen to the study area, which is less urbanized. Properties of the study area were determined based on analysis of elevation, land use and population in QGIS (Figure 1). Rotterdam Centre has a surface area of 4.9 km2, Noord 5.4 km2, Delfshaven 6.0 km2 and Kralingen 7.5 km2, resulting in a total area of 23.8 km2. Terrain elevation varies from 2 m below to 8 m above MSL for district Delfshaven, 2.6 m below to 2.7 m above MSL for district Noord, 1.9 m below to 4.8 m above MSL for district Rotterdam Centre and 3.0 m below to 5.5 m above MSL for district Kralingen. Kralingen is the most low-lying district, with 87% of its surface area lying below sea level, followed by Noord with 79%, Delfshaven with 49% and Rotterdam Centre with 36%. The districts are heavily urbanized, as 86% of Rotterdam Centre's surface area is paved, 80% of North's surface area and 83% of Delfshaven surface area. Kralingen is less urbanized, as 65% of its surface is paved. Delfshaven, Noord and Kralingen are residential districts, while Rotterdam Centre mainly consists of commercial areas. The highest population densities in Rotterdam are found within the study area. In total, 32,915 inhabitants reside in district Rotterdam Centre, 75,425 in district Delfshaven, 51,675 in district Noord and 53,045 in Kralingen (CBS, 2015). Note that the sub-district Kralingse Bos (Kralingen district) was excluded from the study area, as nobody lives here.



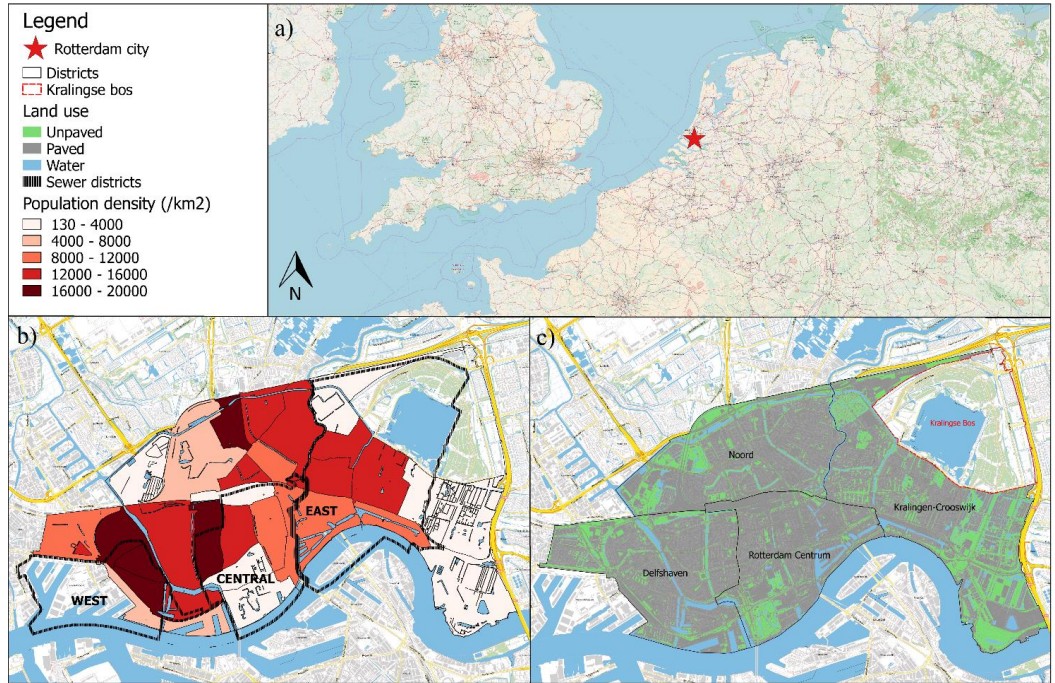

**Figure 1**. (a) Overview of West-Netherlands and the geographic position of the city of Rotterdam. (b) Population density map of the study area and sewer district outlines. (c) Land use map of study area.

As the second biggest city and a major economic center in the Netherlands, Rotterdam city council has a responsibility to make its water system more climate and water robust. The water system consists of an extensive
urban drainage network with numerous assets that require proper management. This includes prevention of surface flooding, which is likely to occur in times of heavy rainfall. Surface flooding is prevented by making use of the multitude of overflow points which are present in the city. However, combined sewer system overflows (CSO's) are detrimental to the water quality of the canals in the city and should only be used as a last resort to prevent flooded streets when no other option is available. A key concept that the Municipality of Rotterdam uses to enhance
water safety in the city is to 'retain, store and drain' water. This concept includes cost-effective multi-functional solutions like the Water Square (Benthemplein), an underground water storage and parking garage (Museumpark) and a 40.000 m$^2$ green roof park (Dakpark Rotterdam), all of which are designed to provisionally store excess storm water during heavy rainfall events (Water Atlas, 2012).

For some years, the Municipality has been working on creating a central automated control system to direct water through the sewer system in order to minimize sewer overflow and flooded streets in the city. In practice, this is still performed through manual control actions, which are necessary during calamities or maintenance in the sewer system. This system consists of different sewer districts. If a sewer district fills up during intense rainfall, two processes can prevent storm water from flooding the streets: overflow pumping (OP) of sewer water into the
Nieuwe Maas river and combined sewer overflow (CSO) into the canals. OP is a control action that can be performed when the sewer system is (nearly) full, in order to prevent CSO or flooded streets. If the discharge capacity of OP and transport capacity of the sewer system are insufficient, sewer overflow on the canals (CSO) occurs. OP has a higher preference than CSO, as CSO has more severe consequences, like deterioration of water quality within the city and malodor. However, only sewer districts that are situated along the Nieuwe Maas river
have                    the                    possibility                    to                    perform                    OP

## 3. Methods

### 3.1 Data pre-processing and software



To perform a temporal correlation analysis a set of 40 events with the largest report and OP values was selected for the analysis of historical flooding data and rainfall. All reports situated in the study area were obtained, corresponding with the selected events. Reports lacking GPS coordinates or a problem description were discarded, as well as reports with irrelevant problem categories. Relevant problem categories were determined as follows:

'Gully pot', 'Sewer', 'Malodor', 'Lack of drainage' and 'Flooding in building'. Eventually, a total set of 8070 reports was obtained for the analyses. OP volumes were corrected for additional inflow originating from other sewer districts and a variety of smaller and larger pumping stations. More information on this can be found in Appendix Table A.1-4. Afterwards, OP volumes of each sewer district were normalized by their contributing area size to convert to unit water column depth (per sewer district) per day. OP normalized depths from the different

districts were then summed up to find total OP values per event. In addition, rainfall values, originating from a single National Rainfall Radar pixel situated at the center of the study area, were selected as a representative overall estimate.

For a spatial correlation analysis, National Rainfall Radar images were processed to produce rainfall raster files for rainfall depth over different durations. This takes into account the effect of spatial heterogeneity of rainfall. A fine

resolution grid of 0.25x0.25 km, corresponding with the typical scale of local flooding events, was used for spatial analyses of urban pluvial flooding relationships. The land use map was converted into imperviousness by assigning imperviousness degrees per land class, resulting in degree of imperviousness per grid cell. First, appropriate runoff coefficients were selected according to the existing land classes of the land use. These classes are as follows: buildings, unpaved, paved, semi paved, water, infrastructure and remaining. 'Buildings' are defined by rooftops.

The 'unpaved' class is defined by green pervious surfaces like vegetation or lawns. The classes 'paved', 'semi paved' and 'infrastructure' all consist of a combination of asphalt or brick pavement surfaces and can be assigned one runoff coefficient. The 'water' class represents open water bodies, which do not contribute to surface runoff. Hence, this class was not included in the imperviousness calculation. The 'remaining' class consists of civil structures like tunnels or bridges and are thus characterized by asphalt or concrete. Degree of imperviousness was

calculated by dividing impervious surface area by total surface area, according to Eq. (2). Impervious surface area was calculated by multiplying land class surface areas by their corresponding runoff coefficient. From ranges of runoff coefficients (Butler and Davies, 2004), the average runoff coefficient was selected per land use class (Table 2).

**Table 2. Overview** of assigned runoff **coefficients based** on pre-defined land **classes**.

| | Land class | | | |
|---|---|---|---|---|
| | (Semi) paved or infrastructure | Buildings | Unpaved | Remaining |
| Runoff coefficient | $C_1$ | $C_1$ | $C_2$ | $C_3$ |
| Value | 0.8 | 0.8 | 0.2 | 0.9 |

$$\text{Imperviousness} = \frac{\sum(Area_{semi\ paved,paved,infrastructure,buildings})*C_1 + \sum(Area_{unpaved})*C_2 + \sum(Area_{remaining})*C_3}{\sum Area_{total}} \quad (2)$$

Here, reports were selected that correspond with the top 40 rainfall events to minimize the number of non-flood related reports. Population density was also computed at grid scale. The influence of elevation differences on

surface runoff was analyzed through flow path analysis in QGIS. The 1x1 m DEM excludes surface areas defined by canals, ponds or rooftops, as they cannot be (directly) linked to surface runoff. The DEM was corrected for local depressions, as they result in local ponds of water with no outflow point. In reality, the water level of these ponds rises until they are connected to a surface water body or neighboring watershed, if present. To correct for this effect,

the DEM was treated with a filling process. Flow accumulation was modelled by the D8 algorithm, which corresponds with Single Flow Direction (SFD). This means that flow from a certain pixel can only flow into one neighboring cell based on slope. Flow was also defined by accumulation through neighboring cells. The correction method of the DEM corresponds with other studies (Gaitan et al., 2015; Olivera and Maidment, 1999; Tarboton et al., 1991).

QGIS was used for spatial and statistical analyses. Python and QGIS were used to handle or merge large numbers of rainfall raster files. In addition, Lizard was used, which is a water information portal that presents a variety of water management data, like rainfall-, elevation- and land use data (Nelen & Schuurmans, 2017a).



### 3.2 Temporal correlation analysis

We investigated the relation of report- and OP data with rainfall intensity on a variable temporal scale. Note that the term 'rainfall intensity' is used to express rainfall depth over a defined duration based on the center National Rainfall Radar pixel. Therefore, rainfall depth over different durations was defined according to Eq. (1):

$$d_t = \frac{p_{max}\,(mm)}{t\,(hour)} \tag{1}$$

where, $p_{max}$ (mm) is the maximum rainfall depth of a given duration $t$ (hour) within the 24 hour duration of an event. For instance, the variables $d_{0.25}$, $d_1$ and $d_{24}$ imply the maximum rainfall depths over every 15 minutes, 60 minutes and 24 hours, respectively. Therefore, 5 minute rainfall depth values (National Rainfall Radar) were accumulated to correspond with these variables. The $d_{24}$ variable directly resulted in the maximum rainfall depth, based on the 24 hour duration of an event. For the $d_{0.25}$ and $d_1$ variables, the maximum depth during an event was selected as a characteristic rainfall value. It follows from other research that sub-daily durations are the most relevant for urban hydrologic response (Blenkinsop et al., 2017; Ochoa-Rodriguez et al., 2015; Romero et al., 2011; Simões et al., 2015). Report and OP data were plotted versus rainfall depth and Spearman's rank correlation coefficients were computed. Spearman's Rank correlation coefficient, $\rho$, is a general measure of correlation for non-linear relations (Taylor, 1990). This type of correlation is the most appropriate to use, as the variables may not be identically distributed and relations between them are likely to be non-linear.

### 3.3 Spatial correlation analysis

A spatial rainfall analysis was performed by accumulating reports within the 1x1 km pixels of the National Rainfall Radar images. This made it possible to investigate spatial correlations between rainfall and flood reports. In this analysis, 6 events with the largest number of reports were compared with the spatial distribution of rainfall maxima over different durations ($d_{0.25}$, $d_1$, $d_{24}$). The events have the following dates: 14-07-2011, 26-08-2011, 13-10-2013, 04-11-2013, 23-06-2016 and 03-10-2016. Including more events was not possible based on the minimum event of 40 reports. The events were also analyzed on a finer resolution grid (0.25x0.25 km). In this way, the effect of spatial aggregation was investigated. In contrast to reports, OP data is too coarse for spatial analysis, as its data is only defined per sewer district. The influence of imperviousness and inhabitant density on report density was also investigated.

Another form of a spatial correlation analysis is given by flow path analysis. For flow path analysis, the following assumptions were made: the sewer system is full, open surface water bodies do not interact with each other and do not overflow into the streets (Diaz-Nieto et al., 2012; Gaitan et al., 2015; ten Veldhuis et al., 2011). The total set of reports (8070) was used in this analysis, as this resulted in the most complete depiction of flooding incidents that are potentially caused by surface runoff. Even on dry days, the report count can still be high due to a preceding day of heavy rainfall. In order to conclude whether surface runoff accumulation contributed to urban pluvial flooding, the correlation between distance to outflow point and nearest neighbor distance between reports were analyzed. As more water accumulates towards the outflow point, the probability of flooding incidents may be higher there. If true, reports are clustered around the outflow point of the delineated urban watersheds. In this case, it could be concluded that elevation differences influence urban pluvial flooding.

In the spatial correlation analysis, all correlation coefficients were calculated according to Pearson, as this made it possible to directly compare results with other research (Gaitan et al., 2015, 2016).

### 3.4 Change point analysis

As the relationship between historical flooding data and rainfall is expected to be non-linear and the urban hydrological system in lowland areas like Rotterdam has an important storage component, a threshold value might be present in flood response. Change point analysis, is a powerful statistical tool to derive whether significant changes occur in relationships between datasets. Even very subtle changes, that might be missed in control charts, are picked up. The appropriate datasets were ranked by event number ($y_i$) to calculate accumulation of change (with respect to the mean), from which significant change points were derived. Each event ($i$) was given an equal rank based on rainfall depth in ascending order. Change accumulation ($s_i$) was calculated according to Eq. (3), where $x_i$ provides the value of the observations (report/OP) and $x_{mean}$ is the sample mean of the observations. As long as there is a change accumulation trend downwards, observation values are generally lower than the mean. A downward trend is eventually followed by an upward trend, indicating that observation values are generally higher than the mean. The point where accumulation of change is furthest away from base (zero), is the estimated change point (Taylor, 2000; Weng et al., 2014). Next, bootstrapping was performed to find a confidence interval for the estimated change points. In the procedure of bootstrapping, the original pairs of observation values ($x_i, y_i$) were





randomly reordered 5000 times, from which a distribution of change points was determined. A 95% confidence interval was then derived, based on the 2.5 and 97.5 percentile of estimated change points.

$$s_i = s_{i-1} + (x_i - x_{mean})$$ (3)

### 3.5 Impact of antecedent rainfall

The effect of antecedent rainfall was investigated based on reports. OP data was not suitable for this analysis, as OP volumes are highly influenced by manual control actions. To investigate the influence of antecedent rainfall, the previous analysis for the 40 largest events of report versus rainfall depth over a 24 hour duration ($d_{24}$) was used. Two types of trend lines were constructed that appear to fit the events best, an exponential and a linear trend line. Event deviations from the trend line (residual values) were correlated to rainfall depth over a 48 hour duration ($d_{48}$).

$d_{48}$ corresponds with the sum of $d_{24,i-1}$ and $d_{24,i}$, where '$i-1$' indicates the preceding day and '$i$' the next day. Secondly, $d_{24,i-1}$ was directly correlated to reports during the event. If antecedent rainfall conditions intensify urban pluvial flooding impacts, the analysis should point out that a preceding day of rainfall helps to explain additional variance in reports.

## 4.                                                                        Results

In this section, results of the correlation analyses, change point and antecedent rainfall analysis are presented. Rainfall- and OP statistics from the temporal analysis can be found in Appendix Table A.1-5, as well as additional explanation about outliers in Appendix Table A.6-7. Rainfall- and report statistics of the spatial analysis can be found in Appendix Table A.8. Note that Pearson correlation coefficients were indicated with '$r$' and Spearman's Rank correlation coefficients with '$\rho$'.

### 4.1 Temporal correlation analysis

Reports and OP data were investigated against rainfall on three temporal scales. Afterwards, the relation between reports and OP was analyzed. First of all, we investigated how the variables $d_{0.25}$, $d_1$ and $d_{24}$ are correlated. For this, Pearson correlation coefficients ($r$) were calculated. Strongest correlation ($r = 0.94$) was found for the relation between $d_{0.25}$ and $d_1$, meaning that these are closely related.

For the temporal correlation analysis of reports and rainfall, reports obtained during the selected events were plotted against maximum rainfall depth in a 15 minute, 60 minute and 24 hour time window ($d_{0.25}$, $d_1$ and $d_{24}$). This can be observed in Figure 2. First, Spearman correlation coefficients ($\rho$) were calculated for the total set of events and the different rainfall variables. $\rho$ is quite similar for all variables, although the strongest correlation was found for $d_1$ ($\rho_{d_{0.25}} = 0.47$; $\rho_{d_1} = 0.50$; $\rho_{d_{24}} = 0.48$).

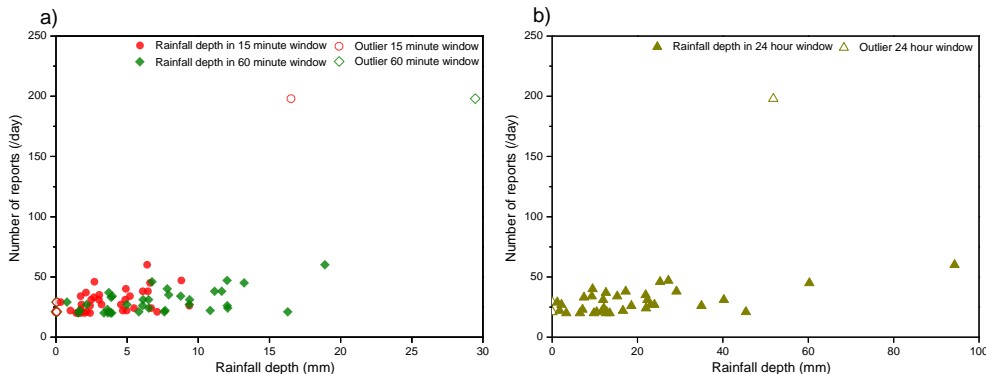

**Figure 2**. (a) Relationship between the maximum rainfall depth (mm) over a duration of 15 minutes, 60 minutes and the number of reports (/day). (b) Relationship between the rainfall depth (mm) over a duration of 24 hours and the number of reports (/day).

To investigate the influence of outliers, significant outliers were identified and filtered out. Four possible outliers were identified: 15-7-2011, 6-6-2016, 23-6-2016 and 24-6-2016. The events of 15-7-2011, 6-6-2016 and 24-6-2016 were selected as there was no rainfall ("Zeros"), but quite some reports (>20 per event). Event 23-6-2016 was




selected as it caused an exceptionally high number of reports (198). The absolute effect of filtering out these events on the correlation coefficients can be observed in Appendix Table A.6. The elimination of 23-6-2016 caused only a slight decrease of $\rho$, as Spearman is little sensitive to events that are situated in the tails of a set of events. Eliminating both the 23-6-2016 and "Zeros" events resulted in the combined effect of lower $\rho$ for all variables

5    ($\rho_{d_{0.25}}$ = 0.40; $\rho_{d_1}$ = 0.44; $\rho_{d_{24}}$ = 0.42). With or without outliers, all correlation coefficients were statistically significant at a 95% confidence level. While event 23-6-2016 has an exceptionally high number of reports, it is not an outlier in the statistical sense, since it is a valid, but rare event. It was therefore relevant to include it in further analysis. Justification of the other proposed outliers (based on rainfall data and logbooks of the city of Rotterdam) can be found in Appendix Table A.7.

10    For the temporal correlation analysis of OP and rainfall, the top 40 OP events were investigated against the rainfall variables $d_{0.25}$, $d_1$ and $d_{24}$, see Figure 3. Based on the full set of the 40 top events of OP, no clear relation  was observed immediately, as there appeared to be some outliers. $\rho$ was initially low and insignificant (95% confidence level) for all variables ($\rho_{d_{0.25}}$ = 0.13; $\rho_{d_1}$ = 0.25; $\rho_{d_{24}}$ = 0.38).

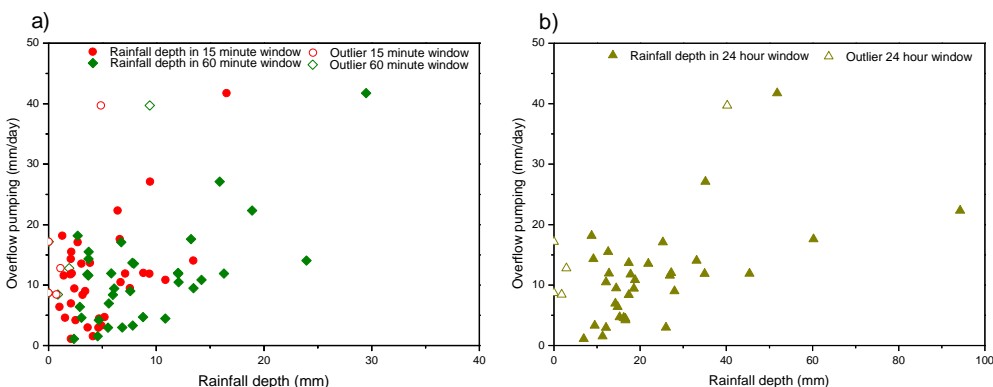

**Figure 3**. (a) Relationship between the maximum rainfall depth (mm) over a duration of 15 minutes, 60 minutes and overflow pumping (mm/day). (b) Relationship between the rainfall depth (mm) over a duration of 24 hours and overflow pumping (mm/day).

Outlying events have two dominant properties: no rainfall and low OP depth; little rainfall and high OP depth. These outliers were respectively event 11-10-2013 (OP 61 mm/day); 10-9-2013 (OP 40 mm/day) and 11-09-2013, 15-07-2011, 22-12-2011, 23-12-2011 (no significant rainfall; "Zeros"). The influence of outliers on the correlation coefficients was investigated by filtering them out (Appendix Table A.6). The elimination of event 11-10-2013

20    resulted in higher Spearman correlation coefficients for all variables ($\rho_{d_{0.25}}$ = 0.19; $\rho_{d_1}$ = 0.32; $\rho_{d_{24}}$ = 0.41).





To identify outliers we investigated the relation between OP depth (mm/day) and rainfall depth ($d_{24}$) by a ratio of OP/rainfall. The ratio would be expected to be below one, as this implies a rainfall depth larger than the OP volume discharged into the Nieuwe Maas river. This can be explained

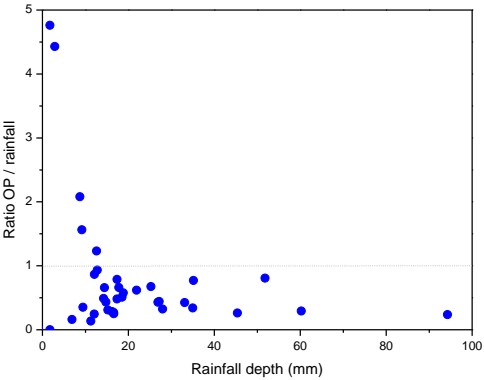

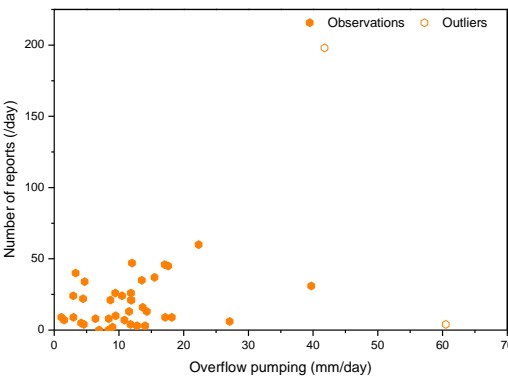

by a part of the rainfall being stored on the streets or

in underground water storages present in the city center and by flow to the waste water treatment plant. If the ratio is above 1, this means that additional water is pumped out of the system, which can be explained by calamities, flushing of canals after CSO's or flow from the canals into the sewer system. A graph that depicts the OP/rainfall ratios is shown in Figure 4. In this graph, the outlying events that were above a ratio of 3 correspond with events 23-12-2011, 11-9-2013, 11-10-2013 that were identified as outliers. The events at the origin of the graph were also

identified as outliers, namely: 15-7-2011 and 22-12-2011. There are two events that are close to a ratio of 2, which were probably caused by the reasons described in this paragraph and were kept for analysis. All of the other entries are generally a little below ratio 1, as expected.

**Figure 4**. Ratio of overflow pumping (OP) depth divided by rainfall depth (24 hour time window) simplifies the identification of outliers, as this ratio should be below 1 for all rainfall depths.

**Figure 5**. Relationship between overflow pumping (OP) (mm/day) and number of reports (/day).

In Figure 5, number of reports for the 40 top events of OP were plotted against OP depth. In this dataset, OP and reports are positively correlated, although no clear relation could be observed. As overflow pumping is used to prevent urban pluvial flooding, it is logical that reports did not increase much within the lower range of OP depth. However, in case of a heavy rainfall event, larger OP depths would have been necessary to prevent urban pluvial flooding. In the higher range of rainfall depths, the probability increases that the OP and transport capacity of the

drainage system cannot cope with the amount of rainfall. Therefore, the probability on a higher number of reports increases. This theory corresponds with the behavior of the events that can be observed in Figure 5, where event 23-6-2016 is statistically not considered an outlier.





## 4.2 Spatial correlation analysis

First, the influence of spatial heterogeneity of rainfall on reports was analyzed. Next, the correlation between imperviousness and report density was investigated. Finally, a flow path analysis was performed to analyze the influence of elevation differences on surface runoff.

For the analysis of spatial heterogeneity of rainfall and reports, the 6 top events were analyzed at a 1x1 km and a 0.25x0.25 km grid. Pearson correlation coefficients ($r$) were calculated based on the relation between the rainfall variables ($d_{0.25}$, $d_1$, $d_{24}$) and report density on the two mentioned spatial aggregation scales. No significant correlations (95% confidence interval) were found for the rainfall variable $d_{24}$. Correlations for the variables $d_{0.25}$ and $d_1$ are statistically significant, from which it followed that rainfall depths over sub-daily durations are the most

essential in explaining the spatial spreading of reports. For these variables, the following Pearson correlation coefficients were found: $r_{d_{0.25}} = 0.41$, $r_{d_1} = 0.44$ for a 1x1 km grid and $r_{d_{0.25}} = 0.22$, $r_{d_1} = 0.23$ for a 0.25x0.25 km grid. The importance of spatial aggregation scale followed from the significant decrease of the correlation coefficients for a finer grid. This decrease can be explained by more zero report density values as well as higher peak values (~multiplication factor 10) for the finer grid. From this, it followed that reports are highly clustered and

that a coarser spatial aggregation resolution has a smoothing effect on report densities.

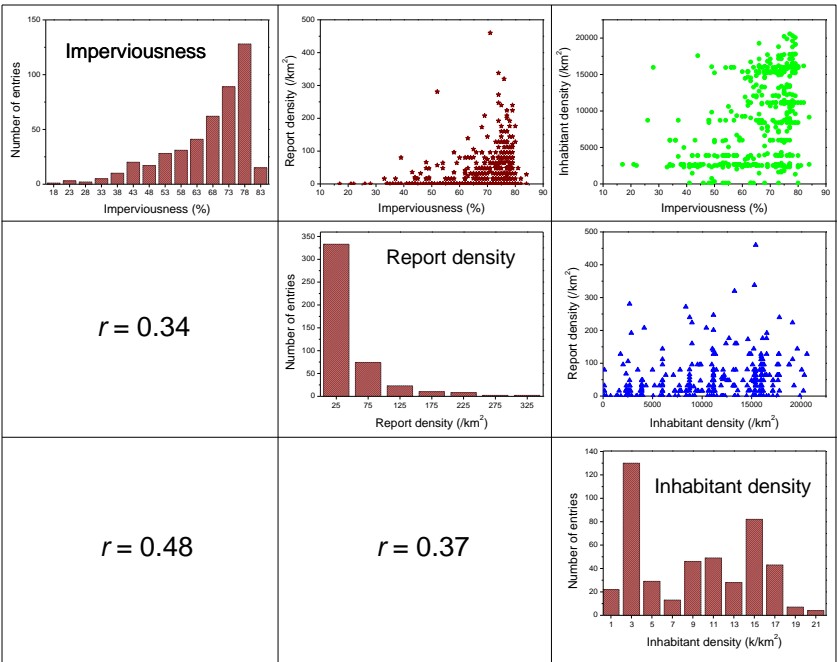

**Figure 6**. Parameter properties and relations of imperviousness, report density and inhabitant density are visualized in a correlation matrix. The diagonal contains the parameters of interest and corresponding histograms. The upper triangle contains scatter plots of parameter relations and the lower triangle shows the corresponding Pearson correlation coefficients ($r$).

Imperviousness was determined per grid cell to correlate it to report density, as explained in Sect. 3.1. The influence of imperviousness on rainfall related reports was investigated based on 991 reports associated with the top 40 rainfall events, which were accumulated within the fine resolution grid. Next, the correlation between the two parameters was calculated. A Pearson correlation coefficient of 0.34 was found, which is statistically significant, meaning there is some explanatory power behind degree of imperviousness as an urban pluvial flooding

parameter. The data did appear to be clustered in the higher range of imperviousness. A change point analysis will be conducted (see Sect. 4.3) to identify whether there is in fact a threshold above which number of reports increases significantly. Since report density and imperviousness are both related to inhabitant density, these correlations were also investigated. The population density map, shown in Figure 1, was converted to the 0.25x0.25 km grid to determine the correlation with report density. A strong correlation between inhabitant density and report density



would imply that most of the spatial variability of reports can be explained by the presence of inhabitants. The value of Pearson correlation is 0.37, meaning weak but statistically significant correlation was found. Nonetheless, population density definitely has some influence on the spatial variability of reports, although it is not a very strong predictor. The correlation between imperviousness and inhabitants was calculated to be 0.48, which means that population density and degree of imperviousness are indeed related. All parameter relations, described in this paragraph, are visualized in Figure 6.





After processing the 1x1 m DEM as indicated in Sect. 3.1, 58 urban catchments or watersheds were delineated, using a threshold value of 300 m$^2$ for minimum size of the watersheds. This size corresponds with flooding on street scale. Since the study area is really flat, overland flow accumulation is only significant at a few locations, which are defined as watersheds. In other words, the areas in between the watersheds are the areas with no

significant overland flow accumulation (< 300 m2). Hence, a flow path analysis for non-watersheds is not necessary. While deriving the watersheds, also a stream map was created, showing the preferential flow paths that are defined by the highest accumulation of surface runoff. The outflow point of an urban watershed is defined by the downstream end of the stream. The urban watersheds were clipped with the total set of reports, resulting in a subset of 251 reports. 82 of these reports had identical locations as other reports from different events. To prevent

duplicate reports from affecting the relation that is under investigation, they were removed before further analysis. Reports were considered duplicates if both location and date are equal, as they can for example be caused by impatient callers. This resulted in a set of 245 reports; 33 of the 58 delineated watersheds contain reports. These 33 watersheds were investigated to find out whether elevation differences can be related to urban pluvial flooding.

Figure 7         shows the
used         DEM, the
delineated         urban
        watersheds,
flow paths         preferential
        and reports.

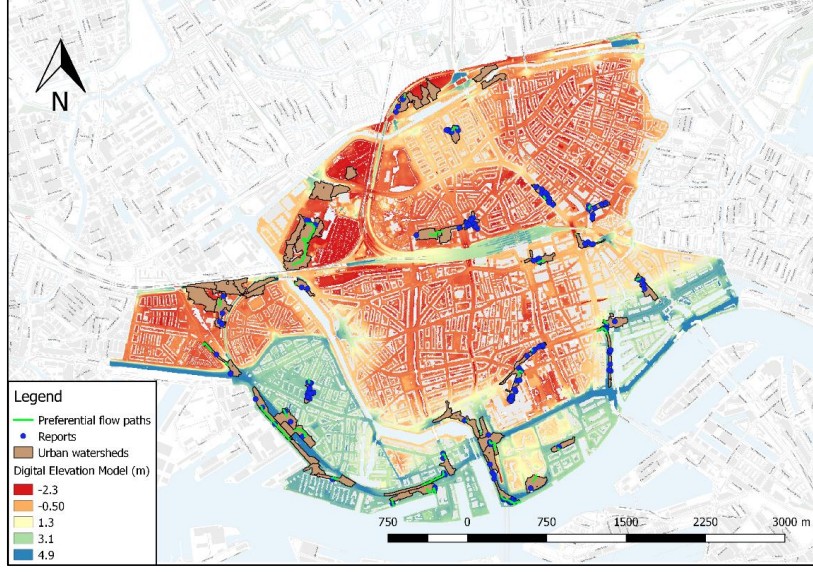

**Figure 7**. This figure shows the Digital Elevation Model (DEM) of the study area and the delineated watersheds, including reports situated in these watersheds. Flow paths can also be observed. The DEM unit is in meters.

First of all, distance to outflow point was determined through flow path length. These flow paths, which start at a report location and end in the watershed outflow point, were derived through the drain function in the GRASS environment of QGIS. To normalize the effect of different watershed sizes or shapes, distance to outflow ratio was

used, which is expressed as $Dist.\,to\,outflow_i/Dist\,to\,outflow_{i,max}$, where the maximum distance is watershed specific. Next, nearest neighbor distances were determined through a distance matrix, resulting in the Euclidian distances between reports. Here, the minimum distance to a neighboring report gives the nearest neighbor distance. The Pearson correlation coefficient that was calculated for the relation between distance to outflow ratio and nearest neighbor is statistically insignificant. There does not appear to be any preference for reports to be clustered around

the outflow point. It followed that redirecting of surface runoff due to elevation differences does not have a significant influence on urban pluvial flooding. This also means that no threshold can be derived considering the distance to outflow ratio in relation to clustering of reports.




### 4.3 Change point analysis

Based on the set of the 40 top events in terms of number of reports, it was possible to derive threshold values above which reports increase rapidly. The derivation of these values was performed for the rainfall variables $d_{0.25}$ and $d_1$, as these sub daily durations are most relevant for urban hydrologic processes. To determine the threshold values in

a statistically robust way, a change point analysis was performed. All 40 observations for the variables $d_{0.25}$ and $d_1$ were ranked based on rainfall depth. Ranking the observations decreases the vulnerability of the analysis to outliers. Change accumulation, as explained in Sect. 3.4, was calculated according to Eq. (3), where largest accumulation of change (point furthest away from zero) indicates the change point. The change accumulation chart that resulted from the originally ordered set of observations, can be seen in Figure 8a. It follows that the point furthest away

from zero has rank 31 for both variables. This observation rank corresponds with a change point value of 5.5 mm/15min and 10.8 mm/60min. A Box-and-Whisker plot of report values before and after the change point is provided in Figure 8b. To increase the robustness of the actual change points, confidence intervals and bootstraps were used. 5000 bootstraps were generated to create a distribution of change points and a 95% confidence interval was used to address the change point probability. This resulted in the following change point ranges: 2.2-9.4

mm/15min and 4-19 mm/60min. Taking the average as the best guess of the actual change point, the following thresholds were found: 5.7 mm/15min and 10.5 mm/60min. If these rainfall depth-duration values are exceeded, the probability of occurrence of urban pluvial flooding in Rotterdam increases remarkably. A similar procedure was performed, for the subset of 'flooding in building' related reports. The following change point ranges were found: 4.9-9.4 mm/15min and 9-19 mm/60min, for a 95% confidence interval. This leads to average change points of 6.6

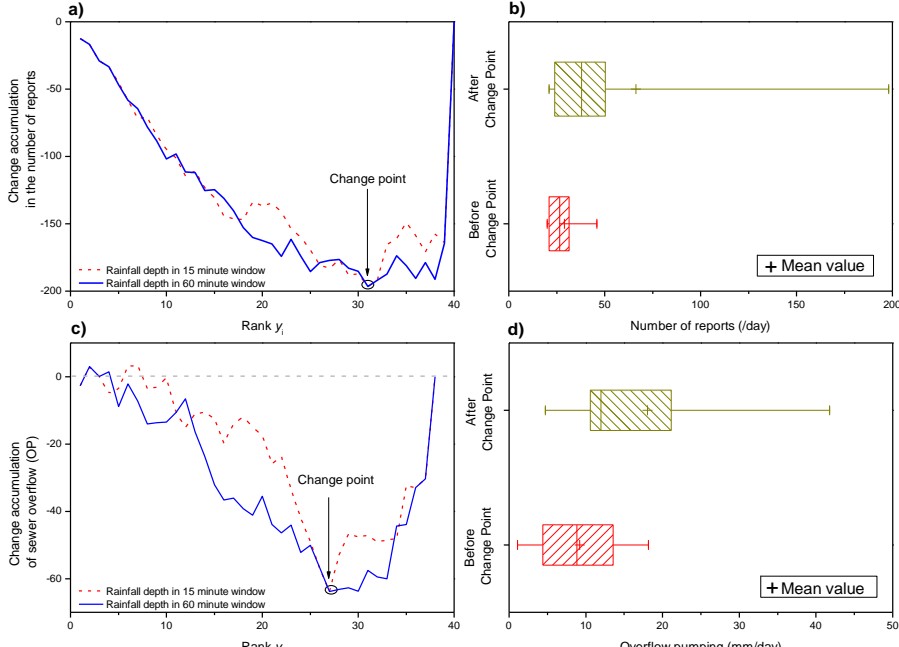

mm/15min and 12.5 mm/60min.



**Figure 8**. (a) Change accumulation chart of report and rainfall analysis, indicating the change point. (b) Box-and-Whisker plot of report values before and after the change point. (c) Change accumulation chart of OP and rainfall analysis, indicating the change point. (d) Box-and-Whisker plot of OP values before and after the change point.

Thresholds were also derived from OP data for the rainfall variables $d_{0.25}$ and $d_1$. Events 11-10-2013 and 10-9-2013
were discarded, as they are the result of a different process than the relation between rainfall and overflow that we
aim to describe. Again, a change point analysis was performed to determine the threshold values in a statically
robust way. The chart of change accumulation can be observed in Figure 8c. The point farthest away from base has
rank 27 for both variables. This corresponds with the following change points: 5.2 mm/15min and 12.1 mm/60min.
A Box-and-Whisker plot of OP values before and after the change point is provided in Figure 8d. A 95%
confidence interval and 5000 bootstraps were used. Based on the results, there is a 95% probability that the change
points lie between 3.6-9.4 mm/15min and 6-13 mm/60min. Taking the average as the best guess of the actual
change point, the following thresholds were found: 6.0 mm/15min and 11.5 mm/60min. Once these rainfall depth-
duration values are exceeded, the probability of experiencing urban pluvial flooding in Rotterdam increases
significantly.

As indicated, a change point analysis was also performed for the relation between imperviousness and report
density. The original set of observations was bootstrapped 5000 times and a 95% confidence interval was taken.
This resulted in a change point range of 66-74% imperviousness and an average change point of 70%. This means
that if this threshold of 70% is exceeded, there is a significant probability of increase in report densities. A
cumulative probability distribution pointed out that 50% of all observations are situated in the range 70-84%
imperviousness, confirming that observations are clustered in the higher range of imperviousness. The mean of the
set of observations in the lower imperviousness range (17-69%) is 18 reports/km$^2$, while it is 59 reports/km$^2$ for the
higher range (70-84%). A Mann-Whitney U test and a Student T-test were both applied to indicate that this
difference in mean of the two subsets is statistically significant (p < 1E-4 for both tests).

## 4.4 Impact of antecedent rainfall

In this section, we investigated whether a preceding day of rainfall can be used to explain additional variance in
reports. The $d_{48}$ values were investigated against residual values, originating from event deviations from an
exponential trend, as can be seen in Figure 9. Residual values were also calculated for a linear trend. Afterwards,
Spearman's Rank correlation coefficients were calculated to find out whether antecedent rainfall explains
additional variance in reports. Based on the exponential trend, a $\rho$ of 0.09 was found. The linear trend resulted in a
$\rho$ of 0.01. It is clear that this is not statistically significant. To finalize this investigation, Spearman's Rank
correlation was directly determined between $d_{24,i-1}$ and the corresponding number of reports (during the events). A
$\rho$ of -0.09 was found, which is again an insignificant correlation. From these results, it again appeared that there is
no clear relation between antecedent rainfall and urban pluvial flooding for a temporal resolution of 24 hours.

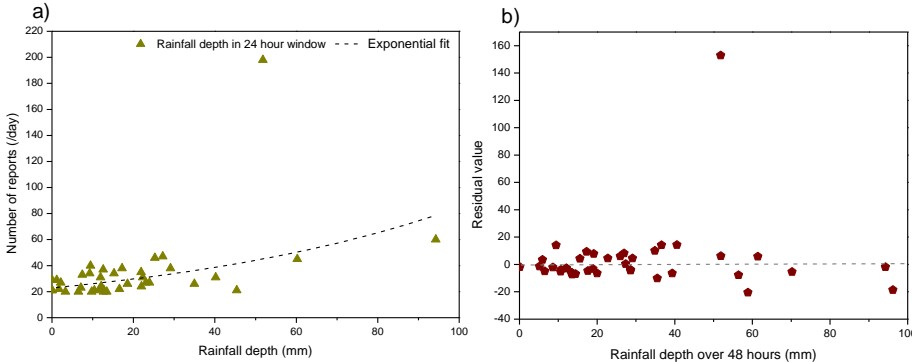

**Figure 9**. (a) Relation between rainfall depth in 24 hour window and number of reports. Also, the exponential trend is displayed. The difference between the report value of an event and the corresponding trend value results in a residual value. (b) Residual values are investigated against rainfall depth over 48 hours ($d_{48}$). No clear relation can be observed.





## 5. Discussion

The temporal correlation analysis showed that reports had the strongest Spearman rank correlation with maximum rainfall depths at a 60 minute duration ($d_1$). However, correlations with maximum rainfall depths at a 15 minute ($d_{0.25}$) and 24 hour duration ($d_{24}$) are only slightly lower. The OP dataset showed strongest correlations with $d_1$ and

$d_{24}$. The $d_{24}$ variable is appropriate to provide information on the filling up of the drainage system as a whole, as it describes a longer time interval. Therefore, as OP is defined by larger temporal- and spatial scale processes, it is to be expected that a high correlation with the $d_{24}$ variable was found. The origin of reports, on the other hand, is dominated by small-scale temporal- and spatial processes, like local blockages of gully pots or sewer pipes, explaining the highest correlation with the $d_1$ variable.

The spatial correlation analysis showed that there is some explanatory power in the spatial variability of rainfall to predict the spatial distribution reports. Strongest correlations were found with the variables $d_{0.25}$ and $d_1$. This confirms the relevance of sub-daily temporal resolutions when investigating the impact of rainfall on urban hydrology (Blenkinsop et al., 2017; Ochoa-Rodriguez et al., 2015; Romero et al., 2011; Simões et al., 2015). It should be noted that relations found are strongly dependent on the spatial aggregation scale. A coarser spatial

aggregation scale has a smoothing effect on variable densities, which resulted in higher correlation values. For example, the relation of spatial variability of rainfall was investigated against report density at two spatial aggregation scales (0.25x0.25 km and 1x1 km). A factor of 2 difference in correlation value was found between these two scales, meaning that the 1x1 km scale resulted in Pearson correlation coefficients that are twice as high as at the 0.25x0.25 km scale. A similar conversion factor was found when imperviousness was analyzed against report

density. In this study, a Pearson correlation coefficient of 0.34 was found for the 0.25x0.25 km scale, while a similar study (Gaitan et al., 2016) resulted in a Pearson correlation coefficient of 0.69 based on a 1x1 km scale (for the city of Amsterdam). For the flow path analysis, a similar result as other research was found (Gaitan et al., 2015), where it followed that reports are not clustered around the outflow point of the urban watersheds (for the Kralingen district of Rotterdam). However, the results of this analysis might have been influenced by the fact that report

locations can be biased towards the presence of buildings and the corresponding address of the person who reports. In this case, reports are not the best measure to address the relation between overland flow and urban pluvial flooding.

The change point analyses, based on reports, OP and imperviousness, have led to the identification of threshold values. These threshold values of 5.7 mm/15min, 10.5 mm/60min (based on reports) and 6 mm/15min, 11.5

mm/60min (based on OP) resulted in average thresholds of 6 mm/15min, 11 mm/60min, after bootstrap analysis. Traditionally, Dutch sewer systems were designed to have a transport capacity up to 20 mm/60min (Amsterdam Rainproof, 2017; RIONED, 2015). This design standard has also been implemented for Rotterdam's drainage system. In reality, the functioning of the drainage system is often far from optimal. In some cases, e.g. due to in-sewer defects or bad maintenance, this drainage capacity could be greatly reduced. If it is assumed that the

functioning of a drainage system would be reduced up to 50%, this means the drainage system should function properly up to 10-20 mm/60min of rainfall. Therefore, the found hourly threshold value of 11 mm/60min, which is partially based on in-system defects or bad maintenance (reports), is in the right order of magnitude. The 15 minute duration threshold value was found to be around 6 mm. The explanation behind this is found in the high (Pearson) correlation (0.94) for the relation between the variables $d_{0.25}$ and $d_1$. In fact, the average ratio of $\frac{mm/60min}{mm/15min}$ based on

the 40 events of both the report- and OP analysis is about 2. This means that a threshold of 6 mm/15min corresponds, on average, with 12mm/60min of rainfall, which is line with the hourly threshold value. The average thresholds found for the 'flooding in building' report category were 6.6 mm/15min and 12.5 mm/60min. It is logical that these thresholds are a little bit higher than the ones from the full set of reports, as higher rainfall amounts are more likely to cause damage to buildings. The found hourly threshold value for 'flooding in building'

is similar to the threshold of 10 mm/60min found in similar research based on insurance claim data associated with flooding from streets into private properties (Spekkers et al., 2015),

The 70% (±4 %, 95% confidence level) imperviousness threshold derived from the change point analysis indicates that there is a critical value of imperviousness, above which reports increase significantly. While this threshold is

relevant for the city of Rotterdam, it might be different for other cities, as the functioning of the drainage system also plays a dominant role in the occurrence of urban pluvial flooding. It should also be noted that the absolute threshold value is a product of the land class map of the city of Rotterdam and the selected runoff coefficients.





Therefore, the outcome that there is indeed a change point that implies a higher probability of urban pluvial flooding is more valuable than the exact figure. Hence, imperviousness was confirmed to be an appropriate parameter to predict urban pluvial flooding.

5 Finally, reports were investigated against antecedent rainfall to find out whether the latter could be used as an urban pluvial flooding predictor. The results have only pointed out insignificant correlation coefficients, meaning antecedent rainfall is not a good urban pluvial flooding predictor in our study. This is in line with other research (Smith et al., 2013), where no strong correlation was found between antecedent rainfall and storm event hydrologic response for a variety of urban catchments.

The findings from this research are dependent on the assumption that all reports provide a reliable representation of the actual flooding incidents that occur during rainfall events. In reality, this might not always be the case for a variety of reasons. Some people might for example not report anymore, as they have reported a problem repeatedly without seeing much improvement. Language might also form a barrier for inhabitants with an immigrant 15 background, meaning they might not report. Besides this, citizens that experience more serious flooding nuisance might call the emergency services instead of the Municipality. Uncertainties cannot be avoided when using crowdsourced data; here they are expected to influence representation, yet observations are considered valid.

## 6. Conclusions

This study was performed to identify urban pluvial flooding predictors or thresholds. Historical flooding data 20 (crowdsourced flooding reports and overflow pumping volumes) have been used as indicators of urban pluvial flooding. This allowed us to perform the study without the need to run hydrodynamic models. The conclusions are summarized as follows:

- Strong correlations were found between flooding and maximum rainfall depth for a 15 minute ($d_{0.25}$) and 60 minute ($d_1$) duration. These variables have been used to derive rainfall threshold values of 6 mm/15min 25 ($\pm$ 3 mm) and 11 mm/60min ($\pm$ 6 mm) from the historical flooding data. These thresholds are potentially valuable as part of an early warning system.
- Crowdsourced flooding reports appear to be clustered in the higher range of imperviousness. A threshold value of 70% ($\pm$ 4 %) imperviousness has been derived, above which the number of reports increases significantly. Therefore, it follows that degree of imperviousness can be used as an urban pluvial flooding 30 predictor.
- From the flow path analysis, it can be concluded that elevation differences in Rotterdam city center do not contribute significantly to urban pluvial flooding, as reports are not clustered around watershed outflow points.
- Finally, it has been found that antecedent rainfall is not useful in improving urban pluvial flooding 35 predictions, based on a temporal resolution of a day.

For future research, it would be highly valuable to extend the dataset to include more high rainfall intensity events. Also, performing a more in-depth analysis in which a clear distinction between in-system defects and system exceedance can be made, could lead to further insights into urban flood prediction.

*Data availability.* Rainfall, overflow pumping and report statistics used in the analyses are made available in a 40 separate Appendix.

*Author contributions.* CB and MV designed the study. MS provided statistical analysis methods. CB performed the analyses. JS helped to collect the data. CB wrote the draft and MV, MS and XT contributed to interpret the results and to refine the paper.

*Competing interests.* The authors declare that they have no conflict of interest.

45 *Acknowledgements.* We would like to thank the Municipality of Rotterdam for providing us with their water system data and supporting us in the performance of the analysis. In particular we want to thank Jerôme Schepers and Elijan Bes for this. We would also like to thank Cees-Anton van den Dool and Lex van Dolderen from water management consultancy Nelen & Schuurmans for providing us with rainfall data based on their National Rainfall Radar product. This work is a contribution to the MUFFIN project (Multi Scale Urban Flood Forecasting) through 50 funding by Water JPI and Dutch National Science foundation (NNO) provided to the third author.





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



Appendix A

**Table A.1. Sewer district characteristics.**

| Sewer district | Area size (m²) | Pump discharge (m³/h) | Discharge into |
|---|---|---|---|
| 1 | 600400 | 1270 | Central district |
| 6 | 1217500 | 2475 | Central district |
| Center (9) | 1834900 | 2160 | Nieuwe Maas (OP) |
| East (10) | 2338700 | 7200 | Nieuwe Maas (OP) |
| West (11) | 547300 | 2100 | Nieuwe Maas (OP) |
| 13 | 1089400 | 2260 | Central district |

**Table A.2. Pumping station characteristics.**

| Pumping station | Max. pump discharge (m³/h) | Discharge into |
|---|---|---|
| 367 | 90 | West district |
| 399 | 13 | West district |
| 412 | 90 | West district |
| 413 | 24 | West district |
| 415 | 240 | West district |
| 416 | 550 | West district |
| 224 | 15 | East district |
| 361 | 94 | East district |
| 362 | 30 | East district |
| 365 | 144 | East district |
| 368 | 1.6 | East district |
| 369 | 1.6 | East district |
| 1498 | 40 | East district |
| 1604 | 20 | East district |
| 385 | 180 | East district |

**Table A.3. Time of overflow pumping (OP) in hours.**

| Date | Central | East | West |
|---|---|---|---|
| 28-02-10 | 2.6 | 1.3 | 0.0 |
| 02-05-10 | 4.2 | 0.1 | 2.6 |
| 08-06-10 | 2.9 | 0.7 | 0.0 |
| 16-08-10 | 5.7 | 2.8 | 2.7 |
| 13-01-11 | 3.2 | 2.7 | 5.6 |
| 14-07-11 | 12.5 | 2.2 | 5.1 |
| 15-07-11 | 3.1 | 1.0 | 2.7 |
| 26-08-11 | 1.2 | 1.4 | 0.8 |
| 22-12-11 | 6.4 | 0.0 | 5.7 |
| 23-12-11 | 8.6 | 0.0 | 2.6 |
| 11-06-12 | 5.4 | 2.2 | 4.1 |
| 23-12-12 | 0.0 | 0.9 | 8.9 |
| 27-07-13 | 3.5 | 1.9 | 3.4 |
| 09-09-13 | 0.0 | 1.2 | 2.0 |
| 10-09-13 | 13.2 | 3.0 | 13.2 |
| 11-09-13 | 5.6 | 0.7 | 1.6 |
| 11-10-13 | 21.5 | 9.6 | 18.2 |
| 13-10-13 | 6.9 | 3.4 | 4.9 |
| 04-11-13 | 0.0 | 3.1 | 7.8 |
| 09-11-13 | 0.0 | 1.1 | 3.2 |
| 28-07-14 | 1.5 | 0.0 | 5.3 |
| 22-08-14 | 1.6 | 1.1 | 1.2 |
| 08-01-15 | 2.7 | 1.1 | 5.3 |
| 20-02-15 | 2.6 | 0.0 | 4.6 |
| 29-03-15 | 0.0 | 1.2 | 3.9 |
| 14-08-15 | 1.3 | 2.1 | 6.1 |
| 25-08-15 | 0.0 | 1.5 | 1.7 |
| 26-08-15 | 2.8 | 1.0 | 3.2 |
| 27-08-15 | 0.5 | 2.2 | 0.0 |
| 04-09-15 | 6.1 | 0.6 | 10.7 |
| 05-09-15 | 0.0 | 2.2 | 0.0 |
| 17-09-15 | 0.0 | 2.0 | 1.9 |
| 30-11-15 | 0.0 | 2.2 | 2.6 |
| 07-01-16 | 1.9 | 0.9 | 4.9 |
| 30-01-16 | 1.7 | 1.4 | 4.8 |
| 04-06-16 | 1.9 | 0.7 | 4.2 |
| 16-06-16 | 4.2 | 1.1 | 4.7 |
| 20-06-16 | 0.7 | 2.3 | 3.8 |
| 23-06-16 | 21.8 | 6.0 | 9.7 |
| 03-10-16 | 12.3 | 2.0 | 0.0 |





**Table A.4a. Overflow pumping data of temporal analysis**

| Event dates | $d_{0.25}$ (mm/15min) | $d_1$ (mm/60min) | $d_{24}$ (mm/24h) | East (m³/day) | Central (m³/day) | West (m³/day) | Total (m³/day) | Total (mm/day) |
|---|---|---|---|---|---|---|---|---|
| 28-2-2010 | 3.6 | 6.8 | 26 | 1 370 | 7 684 | 0 | 9 054 | 3.0 |
| 2-5-2010 | 3.4 | 7.6 | 28 | 2 216 | 468 | 2 633 | 5 317 | 9.0 |
| 8-6-2010 | 4.6 | 5.5 | 12.1 | 1 546 | 4 004 | 0 | 5 550 | 3.0 |
| 16-8-2010 | 7.1 | 16.3 | 45.4 | 2 993 | 16 756 | 2 722 | 22 472 | 11.9 |
| 13-1-2011 | 2.1 | 3.7 | 12.6 | 1 690 | 16 098 | 5 686 | 23 474 | 15.5 |
| 14-7-2011 | 6.4 | 18.9 | 94.3 | 6 557 | 12 984 | 5 107 | 24 648 | 22.3 |
| 15-7-2011 | 0 | 0 | 0 | 1 640 | 6 171 | 2 719 | 10 530 | 8.7 |
| 26-8-2011 | 4.9 | 7.8 | 9.5 | 633 | 8 393 | 772 | 9 797 | 3.3 |
| 22-12-2011 | 0 | 0 | 0 | 3 385 | 0 | 5 768 | 9 153 | 17.2 |
| 23-12-2011 | 1.1 | 1.9 | 2.9 | 4 525 | 0 | 2 593 | 7 118 | 12.8 |
| 11-6-2012 | 13.4 | 23.9 | 33.1 | 2 843 | 13 034 | 4 102 | 19 979 | 14.1 |
| 23-12-2012 | 1.3 | 2.7 | 8.7 | 0 | 5 310 | 8 934 | 14 244 | 18.2 |
| 27-7-2013 | 10.9 | 14.2 | 18.8 | 1 856 | 11 153 | 3 410 | 16 419 | 10.9 |
| 9-9-2013 | 1.5 | 3.1 | 16.2 | 0 | 7 299 | 1 999 | 9 298 | 4.6 |
| 11-10-2013 | 2 | 2.9 | 15.6 | 6 938 | 18 287 | 13 324 | 38 549 | 39.7 |
| 11-9-2013 | 0.7 | 0.9 | 1.8 | 2 933 | 4 005 | 1 576 | 8 514 | 8.4 |
| 10-9-2013 | 4.9 | 9.4 | 40.2 | 11 322 | 57 832 | 18 368 | 87 523 | 60.5 |



**Table A.4b. Overflow pumping data of temporal analysis**

| Event dates | $d_{0.25}$ (mm/15min) | $d_1$ (mm/60min) | $d_{24}$ (mm/24h) | East (m³/day) | Central (m³/day) | West (m³/day) | Total (m³/day) | Total (mm/day) |
|---|---|---|---|---|---|---|---|---|
| 13-10-2013 | 6.6 | 13.2 | 60.2 | 3 605 | 20 404 | 4 925 | 28 934 | 17.6 |
| 4-11-2013 | 2.7 | 6.8 | 25.3 | 0 | 18 607 | 7 814 | 26 421 | 17.1 |
| 9-11-2013 | 2.1 | 5.6 | 14.2 | 0 | 6 616 | 3 222 | 9 838 | 7.0 |
| 28-7-2014 | 9.4 | 12 | 35 | 800 | 0 | 5 303 | 6 102 | 11.9 |
| 22-8-2014 | 4.7 | 10.8 | 16.6 | 858 | 6 852 | 1 222 | 8 932 | 4.5 |
| 8-1-2015 | 3.1 | 7.9 | 21.9 | 1 438 | 6 727 | 5 315 | 13 480 | 13.5 |
| 20-2-2015 | 1.4 | 3.7 | 26.9 | 1 376 | 0 | 4 667 | 6 043 | 11.6 |
| 29-3-2015 | 3.2 | 6 | 17.4 | 0 | 7 394 | 3 903 | 11 297 | 8.4 |
| 14-8-2015 | 2.1 | 3.7 | 9.2 | 658 | 12 323 | 6 131 | 19 112 | 14.3 |
| 25-8-2015 | 2.5 | 4.7 | 16.6 | 0 | 9 085 | 1 718 | 10 803 | 4.2 |
| 26-8-2015 | 7.6 | 13.5 | 14.5 | 1 487 | 6 200 | 3 249 | 10 936 | 9.5 |
| 27-8-2015 | 4.1 | 4.6 | 11.3 | 253 | 12 962 | 0 | 13 215 | 1.5 |
| 4-9-2015 | 9.4 | 15.9 | 35.1 | 3 194 | 3 662 | 10 784 | 17 641 | 27.1 |
| 5-9-2015 | 2.1 | 2.4 | 6.9 | 0 | 13 053 | 0 | 13 053 | 1.1 |
| 17-9-2015 | 5.2 | 8.8 | 15.2 | 0 | 11 721 | 1 874 | 13 595 | 4.7 |
| 7-1-2016 | 2.2 | 5.8 | 12.8 | 975 | 5 353 | 4 953 | 11 282 | 11.9 |
| 4-6-2016 | 6.7 | 12.1 | 12.1 | 980 | 4 229 | 4 264 | 9 473 | 10.5 |
| 16-6-2016 | 3.9 | 7.8 | 17.4 | 2 226 | 6 326 | 4 719 | 13 270 | 13.7 |
| 20-6-2016 | 2.4 | 6.1 | 18.5 | 378 | 13 876 | 3 841 | 18 095 | 9.4 |
| 23-6-2016 | 16.5 | 29.5 | 51.8 | 11 460 | 35 977 | 9 736 | 57 174 | 41.8 |
| 3-10-2016 | 8.8 | 12 | 27.2 | 6 490 | 12 155 | 0 | 18 645 | 12.0 |





**Table A.5. Reports and rainfall statistics of temporal analysis**

| Event dates | $d_{0.25}$ (mm/15min) | $d_1$ (mm/60min) | $d_{24}$ (mm/24h) | Reports |
|---|---|---|---|---|
| 8-6-2010 | 4.6 | 5.5 | 12.1 | 24 |
| 16-8-2010 | 7.1 | 16.3 | 45.4 | 21 |
| 13-1-2011 | 2.1 | 3.7 | 12.6 | 37 |
| 23-6-2011 | 1.0 | 1.7 | 1.7 | 22 |
| 14-7-2011 | 6.4 | 18.9 | 94.3 | 60 |
| 15-7-2011 | 0.0 | 0.0 | 0.0 | 21 |
| 26-8-2011 | 4.9 | 7.8 | 9.5 | 40 |
| 8-9-2011 | 2.7 | 3.9 | 7.5 | 33 |
| 7-12-2011 | 1.7 | 3.6 | 7.1 | 23 |
| 21-12-2011 | 1.8 | 2.2 | 2.2 | 27 |
| 3-1-2012 | 6.5 | 11.6 | 17.2 | 38 |
| 9-5-2012 | 5.0 | 7.7 | 11.7 | 22 |
| 18-6-2012 | 3.7 | 7.6 | 10.4 | 21 |
| 30-1-2013 | 3.0 | 6.5 | 11.9 | 31 |
| 10-9-2013 | 4.9 | 9.4 | 40.2 | 31 |
| 18-9-2013 | 1.4 | 1.6 | 3.3 | 20 |
| 13-10-2013 | 6.6 | 13.2 | 60.2 | 45 |
| 14-10-2013 | 0.3 | 0.8 | 1.2 | 29 |
| 4-11-2013 | 2.7 | 6.8 | 25.3 | 46 |
| 8-1-2014 | 2.0 | 3.9 | 6.5 | 20 |
| 28-7-2014 | 9.4 | 12.0 | 35.0 | 26 |
| 22-8-2014 | 4.7 | 10.8 | 16.6 | 22 |
| 26-8-2014 | 2.5 | 6.1 | 22.3 | 31 |
| 8-1-2015 | 3.1 | 7.9 | 21.9 | 35 |
| 31-3-2015 | 1.6 | 3.4 | 12.7 | 20 |
| 17-9-2015 | 5.2 | 8.8 | 15.2 | 34 |
| 23-9-2015 | 4.6 | 9.4 | 23.1 | 27 |
| 7-1-2016 | 2.2 | 5.8 | 12.8 | 21 |
| 15-1-2016 | 1.7 | 4.0 | 9.3 | 34 |
| 15-4-2016 | 1.7 | 3.7 | 13.5 | 20 |
| 4-6-2016 | 6.7 | 12.1 | 12.1 | 24 |
| 6-6-2016 | 0.0 | 0.1 | 0.1 | 21 |
| 14-6-2016 | 3.2 | 5.0 | 23.9 | 27 |
| 17-6-2016 | 5.5 | 6.5 | 22.0 | 24 |
| 20-6-2016 | 2.4 | 6.1 | 18.5 | 26 |
| 23-6-2016 | 16.5 | 29.5 | 51.8 | 198 |
| 24-6-2016 | 0.0 | 0.0 | 0.0 | 29 |
| 30-6-2016 | 2.4 | 3.8 | 9.7 | 20 |
| 12-7-2016 | 6.1 | 11.1 | 29.1 | 38 |
| 3-10-2016 | 8.8 | 12.0 | 27.2 | 47 |

**Table A.6. Spearman correlation coefficients.**

| | Analysis | | $d_{0.25}$ (mm/15min) | $d_1$ (mm/60min) | $d_{24}$ (mm/24h) |
|---|---|---|---|---|---|
| Total set | Reports | & | 0.47 | 0.5 | 0.48 |
| | rainfall | | | | |
| | OP | & | 0.13 | 0.25 | 0.38 |
| | rainfall | | | | |
| | OP | & | | | 0.26 |
| | reports | | | | |
| NO 11-10-2013 | OP | & | 0.19 | 0.32 | 0.41 |
| | rainfall | | | | |
| | OP | & | | | 0.33 |
| | reports | | | | |
| NO 23-6-2016 | Reports | & | 0.43 | 0.47 | 0.43 |
| | rainfall | | | | |
| | OP | & | | | 0.21 |
| | reports | | | | |
| NO "Zeros" | Reports | & | 0.45 | 0.49 | 0.46 |
| | rainfall | | | | |
| NO 23-6-2016 & "Zeros" | Reports | & | 0.4 | 0.44 | 0.42 |
| | rainfall | | | | |
| NO 11-10-2013 & 23-6-2016 | OP | & | | | 0.28 |
| | reports | | | | |
| NO 11-10-2013 & 10-9-2013 | OP | & | 0.16 | 0.3 | 0.36 |
| | rainfall | | | | |
| NO 11-10-2013 & "Zeros" | OP | & | 0.25 | 0.41 | 0.52 |
| | rainfall | | | | |
| NO 11-10-2013 & "Zeros" & 10-9-2013 | OP | & | 0.23 | 0.4 | 0.48 |
| | rainfall | | | | |





**Table A.7. Explanation behind outliers.**

| Outliers | Analysis | Explanation |
|---|---|---|
| 15-7-2011 | OP | Max. 19 mm/60min & 94 mm/24h of rainfall on preceding day, continued overflow on next day |
| 15-7-2011 | Reports | Max. 19 mm/60min & 94 mm/24h of rainfall on preceding day, still reports received on next day |
| 22-12-2011 | OP | 2 day calamity in district 9 & 10 concerning pump & pressure pipe malfunction |
| 23-12-2011 | OP | *see 22-12-2011 |
| 10-9-2013 | OP | Malfunctioning pumps; hourly max. rainfall depth < 10mm/60min, while in total 40 mm/24h |
| 11-9-2013 | OP | Still overflow due to occurrences on preceding day (10-9-2013) |
| 11-10-2013 | OP | 16 mm/24h rainfall; extreme rainfall forecast 2 days later, more overflow to create additional storage |
| 6-6-2016 | Reports | Preceding rainfall event, exceeded storage capacity (max. 12 mm/60min, 4-4-2016) |
| 24-6-2016 | Reports | Preceding rainfall event, exceeded storage capacity (max. 30 mm/60min, 23-6-2016) |