# Peer review of "Towards identification of critical rainfall thresholds for urban pluvial flooding prediction based on crowdsourced flood observations"

_Hydrology and Earth System Sciences, 2017_

## Referee Comment (RC1) · Anonymous Referee #1 · 8 Feb 2018

It is not clear the novelty/contribution of the work. The methods are not particularly innovative, nor the research questions. The paper reads somewhat like a consultancy report. While it makes a good case study, I am not sure if it is suitable as a scientific publication.

Does the novelty/contribution lie in the use of crowdsourced data? If so, the paper is not written as if it is. If the novelty is indeed in the use of crowdsourced data, then the paper should focus on the crowdsourced data and make more of a discussion/examination of the use of the data to make a strong case of its associated difficulties/advantages. As it is, the introduction and lit review are only in a general sense with no focus on crowdsourcing. And only about 1/3 of the methods and results are based on crowdsourced data.

Also, I am not sure if the historical complaint reports used can be called "crowd-sourced". The reports are in the order of 10-100 per day (as inferred by Fig. 2). However, in today's context, crowdsourcing commonly refers to sourcing from a large pool of people using the Internet, smartphones, surveillance cameras etc. to obtain observations in the hundreds to thousands to millions. Thus, it may be misleading to describe the observations referred to in this paper as crowdsourced.

Other concerns:

There could be a temporal resolution mismatch between the flood complaint reports used and a storm event. That is, in a storm event, it may not be possible to determine which reports coincide with the peak rainfall. Thus, there are some uncertainties in the "crowdsourced" observations that are nontrivial and may affect the validity of the methods/results.

The paper found a strong correlation between surface imperviousness and the number of flooding reports and concluded that "there is some explanatory power behind degree of imperviousness as an urban pluvial flooding parameter". However, the correlation is likely a spurious correlation due to the correlation between population density and the number of flooding reports, and population density and imperviousness. Thus, this conclusion of the paper is not well-justified. The may be advisable for the authors to use other methods (e.g. multivariate linear regression) to exclude the effects of population density and arrive at a more justifiable conclusion.

Other minor points: 1. Page 4, Line 7, km2 instead of km2 should be used. 2. Page 5, Line 10, I guess the area of the green roof park should be 40,000 m2 instead of 40.000 m2. 3. Page 6, Line 25, equation (1) should be appeared before equation (2). 4. The authors should also remove duplicated flooding reports in their temporal and spatial correlation analysis. 5. Instead of using the rainfall intensity at the center of the

study area for the temporal correlation analysis, I would suggest the authors to use the aerial average rainfall intensity of the whole study area. This is to take into account the rainfall spatial variability, which could be rather high at the scale of the study area. 6. What is the unit of dt, mm or mm/h? It seems to be mm/h according to the definition in equation (1), but is shown to be mm in Figures 2, 3, and 4. The authors should also provide more clarified explanations for the variables in equation (1).

———————————————————

---

## Referee Comment (RC2) · Anonymous Referee #2 · 22 Feb 2018

The prediction of urban flooding (location of occurrence and intensity) is an increasingly relevant topic in the field of urban water management. The aspect is interesting not only against the background of a possibly growing risk of urban flooding due to increasing urbanization on the one hand side and a less rapid underground infrastructure development that has difficulties in keeping up with the aboveground urban growth on the other hand. The fact of potentially changing characteristics of extreme rainfall due to a changing climate further contributes to this significance. The present manuscript discusses i) the prediction of urban flooding solely based on detailed rainfall information and ii) the influence of few catchment characteristics (imperviousness, elevation difference) through correlation analyses for a specific case study in the Netherlands.

[Figure]

*Main points:

1. Setting and boundary conditions of the study are very case specific (flat catchment, specific drainage infrastructure and operational regime (OP), rare data availability (OP data, flood reports)). Hence, transfer of findings to other cases - without having carried out similar analyses for other systems - would be, at least questionable. This clearly lowers the scientific significance of the work (reproducibility). Despite the fact that various interesting aspects of general relevance are discussed (use of citizen-reported flood incidents as ground truth data for urban flooding, changepoint analysis), identified (non-)correlations as well as rainfall thresholds – which relate to the key research questions in the paper - are exclusively valid for the particular case Rotterdam. I am wondering why this clearly limiting aspect has not been discussed in the manuscript. I strongly encourage the authors to address this aspect adequately, e.g. by clearly labeling results and findings as case specific (title, abstract, conclusions) and discussing the relevance of findings for other systems.

2. The title prominently suggests research on a currently popular topic: the collection and evaluation of crowdsourced data to extract meaningful information. The main text then reveals that 'crowdsourced data' are here understood as structured recordings on flood incidents reported from the public (!?) which the researchers "obtained" from an existing database (!?) – cf. page 3, line 39 -41. It is not entirely clear to me if the term "crowdsourced" refers to the fact that different people, i.e. the crowd reported the incidents or that the recordings are received from various different sources but are then formalized and archived. In a way this issue is somewhat peculiar since in previous publications the same data set had been named "citizen-reported flood incidents"... Irrespective of the fact if data used in this study can be referred to as "crowdsourced data" the novelty aspect (in the current version of the manuscript) is marginal. Hence I suggest reconsidering the title formulation or a thorough revision of the paper shifting the focus to the use of crowdsourced flood reports as such, e.g. discussing the quality of this source of information.

3. Using rainfall threshold values (which do not account for the spatial structure, moving patterns of storms) to warn for area-specific flooding incidents is a bit far-fetched. From my point of view, the presented results are not convincing enough to allow lumping changing factors (such as downstream drainage system behavior, operational regime, spatial rainfall variability) which eventually influence the degree of flooding and the location at which flooding occurs into a single rainfall threshold value that predicts flooding in a particular urban subcatchment. I suggest using a physically-based dual-drainage model to systematically partition the influence of these aspects and to so put conclusions on a more solid basis.

4. The spatial correlation analysis between interval-specific rainfall depths and reported surface flooding observations leaves me a bit puzzled. Beside the fact that identified correlations are at the very low end in terms of occurrence and significance, the following points are at least debatable:

i) As this particular part of the analysis is based on only six events I am asking how representativeness is ensured. I do not fully understand why the number of reports must be greater than 40 to allow an event to be included in the analysis. No justification is given on what the selection of this threshold is based on, nor a sensitivity analysis is conducted to show how results alter in case more events (with less reports) are selected. In any case it should be questioned to what extend six events provide enough input for a spatial correlation analysis to come to a meaningful conclusion.

ii) Disentangling dependencies: the weak but still existing correlation between population and report density suggests an inherent dependence between the two variables: the more people live in an area, the more reports can be submitted. Hence it should be discussed to what these variables can be independent at all! Depending on the way how flood reports are submitted, normalizing over population density may be a first step to research this aspect. Technically different, but similar with regard to the dependency aspect, the statement "... imperviousness was confirmed to be an appropriate parameter to predict urban pluvial flooding" (p. 17, line 2-3) is somewhat trivial,

i.e. misleading since i) yes, sealed surfaces produce higher surface runoff and ii) solely considered, the degree of imperviousness does allow a prediction of flooding potential – it must be considered in context with other factors. A multivariate analysis approach is recommended.

iii) Considering drainage network capacity constraints: the spatial analysis somehow ignores the fact that urban flooding can substantially be influenced by hydraulic behavior of the actual drainage network (e.g. hydraulic capacity constraints further downstream in the network may lead to manhole overflows). In other words: here flooding is expected to occur right where the rain cell is present, suggesting that the main cause for surface flooding is the pure amount of rainfall at the spot maybe combined with a limited capacity of street inlets. It remains an open issue to what extend the found spatial correlation is influenced/biased through this aspect (still, it is outlined in the outlook for further research – p. 17, line 37) and if this could be a reason indeed for the decreasing correlation when increasing the spatial resolution of data (finer grid). Moreover, this aspect is very likely to become more relevant when researching systems with higher terrain elevation variability, i.e. elevation difference in the catchment.

*Some minor points (not complete):

- The paper's layout is corrupt at many points, it seems that the manuscript had been submitted in a rush.

- Scatter plots, especially Fig. 2, 3, 5, are difficult to read. In particular outliers are difficult to spot.

- The discussion of particular events in the text referring to Fig. 4, 5 (p. 10, line 10, 22) is useless unless it is indicated in the graph and has a particular meaning.

- The treatment of outliers in the OP data is occasionally fuzzy (p. 9, line 5ff; p. 10, line 10-11) and sometimes arbitrary (p. 15, line 4-6). The authors should revise the analysis to avoid the impression that leaving out particular data was done to let results

look a bit better.

---

## Referee Comment (RC3) · Anonymous Referee #3 · 27 Feb 2018

*** General comments ***

The paper brings new ideas to the understanding pluvial flooding from an urban hydrology perspective. They use crowd sourced data in a good way, where no other data is readily available. The paper is well-structured and well-written with a fluent and precise language. The figures are in general very good – clear and easy to interpret. In general, I am positive to the paper. I hope that the following comments can help to improve it. Especially there are a few things that could be better discussed (see specific comments).

Most of the studies mentioned in the discussion section are written by the authors. It

would be good if you could include references to others work as well and discuss the differences? Preferably studies about pluvial flooding from similar climatic zone, like the UK, Germany, Scandinavia, etc. There might also be relevant work from the U.S.

\*\*\* Specific comments \*\*\*

Abstract. Only rainfall mentioned. Imperviousness and distance to outflows are also investigated.

Sec1. Nice introduction.

Sec2.1. There are only 7 years of data. Did you capture any extreme events? What are the return level of the most extreme rainfall in the study?

Fig1/Sec2.2. Why do the study areas not follow the sewer districts? Probably obvious if you know the system...

Sec2.2. Three administrative districts are introduced, with a fourth district for comparison. These districts are not used in the analyses and neither mentioned in the discussion section (except for Kralingen once). I miss the comparison of the four districts. Why are the introduced and then not discussed?

Sec2.2. The key concept of Rotterdam is introduced. OP and CSO are discussed, however not these solutions. Why do you mention them? Do they have any implications for the results in your study? Discuss or delete.

Eq2. Not needed, enough to mention that it is a weighted average.

Sec3.2&3.3. In the introduction, you argue for the importance to analyse varying temporal and spatial resolution. How did you choose resolution? Why 15 min, 1 h, and 24 hours? Why not 4, 8, 12 hours for instance? Did you analyse the drainage system in some way? Describe the process. Similar question for spatial resolution.

Sec3.3. The assumptions for the flow path analysis are described and there are surprising results. I guess you made the analysis because you expected a relation. This

result is barely discussed. Why where there no relation? Could the elevation be analysed better in a different way? What about distance to the main flow path or size of catchment area upstream instead of distance to downstream outlet? To me it seems strange to analyse the relation between a feature downstream of the flooded area, rather than the upstream area.

Fig7. Confusing with the name "urban watershed". Why are not all the urban area a part of an urban watershed?

Sec3.1. A single National Rainfall Radar pixel is used for each study area. Where are these pixels used? In 4.1 temporal correlation analysis? There is no discussion about how representative one pixel are for the area. Any tests made to ensure this? Discuss shortly.

Sec3.4. It took long time to understand how you did the analyses on change point. This section could be written better. It says "The appropriate dataset were ranked by event number". Don't you mean that it was ranked by e.g. rainfall volume and then given an event number?

Sec4.1. The removal of outliers needs to be better discussed. How do define an outlier? The theory behind seems vague.

Sec4.1. How can you get flood reports on days with no rainfall (Zeros)? Explain the registration. 20 reports during a day with no rainfall seems strange. Are you sure that the radar worked?

Sec4.2. You study imperviousness in the same cell. What about the effect of imperviousness upstream the inundated area? Discuss.

Sec4.2. Impatient callers... How are the registration done? Are all calls registered? What are the reason for someone to call? Do they get compensation for damaged properties? Describe the registration process and discuss the implications for your analyses.

Fig6. Did you use fine or coarse grid for these analyses?

Sec5. Discuss limitations of the study. And as mentioned before, related it to others work from similar cities in the same climatic zone.

Sec5. Spekkers 2015: You write that you found similar results before. Mention the differences between the two studies. Why did you get 7-8 mm/h in that study and 12.5 mm/h in this study? Mention the differences in data and methods used.

*** Technical corrections ***

Fig1a. No need for all the details on the map, show a few city names instead. However, this is not crucial to change. Fig1b&c. Scale not indicated. Fig1c. Hard to read the names. Fig2a & Fig3a. Hard to differ the marks in black and white print. For colour blind, the figure must be difficult to read at all. Fig2b & Fig3b. Difficult to see the two outliers on the left hand side.

Spell out acronyms the first time (e.g. Dutch KNMI, German DWD and Belgium KMI).

Sec5. paragraph 3. ", as higher rainfall amounts are more likely to cause damage to buildings." Compared to what?

---

## Author Comment (AC1) · 10 Apr 2018

**Reply to interactive comments on: Towards identification of critical rainfall thresholds for urban pluvial flooding prediction based on citizen flood observations, hess-2017-751**

**Reviewer #3**

Comment 3.1
The paper brings new ideas to the understanding pluvial flooding from an urban hydrology perspective. They use crowd sourced data in a good way, where no other data is readily available. The paper is well-structured and well-written with a fluent and precise language. The figures are in general very good – clear and easy to interpret. In general, I am positive to the paper. I hope that the following comments can help to improve it. Especially there are a few things that could be better discussed (see specific comments). Most of the studies mentioned in the discussion section are written by the authors. It would be good if you could include references to others work as well and discuss the differences? Preferably studies about pluvial flooding from similar climatic zone, like the UK, Germany, Scandinavia, etc. There might also be relevant work from the U.S.

Response 3.1:

We thank you for reviewing our manuscript and making constructive comments. We agree with your point. In the new version of the manuscript, we will expand the discussion section including references to similar studies, in particular:

1. Mazzoleni, M., Arevalo, V. J. C., Wehn, U., Alfonso, L., Norbiato, D., Monego, M., ... & Solomatine, D. P. (2018). Exploring the influence of citizen involvement on the assimilation of crowdsourced observations: a modelling study based on the 2013 flood event in the Bacchiglione catchment (Italy). Hydrology and Earth System Sciences, 22(1), 391.

2. Paul JD, Buytaert W, Allen S, et al (2018) Citizen science for hydrological risk reduction and resilience building. Wiley Interdiscip Rev Water 5:e1262.

3. Sadler, J. M., Goodall, J. L., Morsy, M. M., & Spencer, K. (2018). Modeling urban coastal flood severity from crowd-sourced flood reports using Poisson regression and Random Forest. Journal of Hydrology, 559, 43-55.

4. Wang, C., Du, S., Wen, J., Zhang, M., Gu, H., Shi, Y., & Xu, H. (2017). Analyzing explanatory factors of urban pluvial floods in Shanghai using geographically weighted regression. Stochastic Environmental Research and Risk Assessment, 31(7), 1777-1790.

5. Cherqui, F., Belmeziti, A., Granger, D., Sourdril, A., & Le Gauffre, P. (2015). Assessing urban potential flooding risk and identifying effective risk-reduction measures. Science of the Total Environment, 514, 418-425.

6. Liu Y, Piyawongwisal P, Handa S, et al (2011) Going beyond citizen data collection with mapster: A mobile+cloud real-time citizen science experiment. Proc - 7th IEEE Int Conf e-Science Work eScienceW 2011 1–6 .

7. Buytaert W, Zulkafli Z, Grainger S, et al (2014) Citizen science in hydrology and water resources: opportunities for knowledge generation, ecosystem service management, and sustainable development. Front Earth Sci 2:1–21 . doi: 10.3389/feart.2014.00026

8. Muller CL, Chapman L, Johnston S, et al (2015) Crowdsourcing for climate and atmospheric sciences: Current status and future potential. Int J Climatol 35:3185–3203 .

9. Herman Assumpção T, Popescu I, Jonoski A, Solomatine DP (2017) Citizen observations contributing to flood modelling: opportunities and challenges. Hydrol Earth Syst Sci Discuss 1–26 .

10. Starkey E, Parkin G, Birkinshaw S, et al (2017) Demonstrating the value of community-based ("citizen science") observations for catchment modelling and characterisation. J Hydrol 548:801–817 .

11. Yang P, Ling Ng T (2017) Gauging through the Crowd: A Crowd-Sourcing Approach to Urban Rainfall Measurement and Stormwater Modeling Implications. 7:553–572 .

Comment 3.2

Abstract. Only rainfall mentioned. Imperviousness and distance to outflows are also investigated.

Response 3.2

We agree.

We will add findings with respect to the impact of imperviousness and the distance to outflow to the abstract text.

Comment 3.3

Sec2.1. There are only 7 years of data. Did you capture any extreme events? What are the return level of the most extreme rainfall in the study?

Response 3.3

The citizen call center system of Rotterdam started to record phone calls since 2010 so we only have collected data for seven years.

We will add specific information describing the rainfall intensity of different return periods, which are 9, 11, 18 and 21 mm for a return period of 1 year, 2 years, 10 years and 20 years respectively. More detailed information about the rainfall will be described in a new table in Section 2.

Comment 3.4

Fig1/Sec2.2. Why do the study areas not follow the sewer districts? Probably obvious if you know the system...

Response 3.4:

From a practical perspective, we focus on the city center of Rotterdam which is determined by the border of the administrative districts. The main reason is that each administrative district is managed by a individual water sector so data are also collected per administrative district. However, since the sub-catchments are relatively small-sized and flat, most of which are in a scale of 5 squared kilometers or less with multiple outflow points. The difference between an administrative district and a sewer district is not very dominating. We will clarify this point in Section 2.

Comment 3.5

Sec2.2. Three administrative districts are introduced, with a fourth district for comparison. These districts are not used in the analyses and neither mentioned in the discussion section (except for Kralingen once). I miss the comparison of the four districts. Why are the introduced and then not discussed?

Response 3.5

We agree with your comment.

In the new version of the manuscript, we will describe three districts just as the city center of Rotterdam without unnecessarily emphasizing any specific district names.

Comment 3.6

Sec2.2. The key concept of Rotterdam is introduced. OP and CSO are discussed, however not these solutions. Why do you mention them? Do they have any implications for the results in your study? Discuss or delete.

Response 3.6

We agree.

They will be removed.

Comment 3.7

Eq2. Not needed, enough to mention that it is a weighted average.

Response 3.7

We agree.

Equation 2 is not needed and will be removed.

Comment 3.8

Sec3.2&3.3. In the introduction, you argue for the importance to analyse varying temporal and spatial resolution. How did you choose resolution? Why 15 min, 1 h, and 24 hours? Why not 4, 8, 12 hours for instance? Did you analyse the drainage system in some way? Describe the process. Similar question for spatial resolution.

Response 3.8

We agree that different temporal or spatial resolutions may lead to different results. For the temporal resolution, we follow what was used in (Spekkers et al. 2017). For the spatial resolution, 1 km was adopted by the resolution of radar rainfall data while 250 m was used as a typical scale of a local street flooding (as explained in section 3.1).

Comment 3.9

Sec3.3. The assumptions for the flow path analysis are described and there are surprising results. I guess you made the analysis because you expected a relation. This result is barely discussed. Why where there no relation? Could the elevation be analysed better in a different way? What about distance to the main flow path or size of catchment area upstream instead of distance to downstream outlet? To me it seems strange to analyse the relation between a feature downstream of the flooded area, rather than the upstream area.

Response 3.9

This analysis was motivated by the study of (Gaitan et al. 2015), which was conducted based on a single, large flood event. This study aimed to check their findings based on a larger number of events. We will clarify this point in Introduction and Discussion.

Comment 3.10
Fig7. Confusing with the name "urban watershed". Why are not all the urban area a part of an urban watershed?

Response 3.10
We agree.
We will replace the name 'urban watershed' by '(separate) sub-catchment'. Between sub-catchments, there are almost no flows, since these are isolated polder areas. Only limited amounts of wastewater are sometimes pumped from one district to another, and then routed to the wastewater treatment plant.

Comment 3.11
Sec3.1. A single National Rainfall Radar pixel is used for each study area. Where are these pixels used? In 4.1 temporal correlation analysis? There is no discussion about how representative one pixel are for the area. Any tests made to ensure this? Discuss shortly.

Response 3.11
We agree with your comment, which was also raised by other reviewers.
We will extract rainfall data per pixel from radar images and calculate an aerial mean value.

Comment 3.12
Sec3.4. It took long time to understand how you did the analyses on change point. This section could be written better. It says "The appropriate dataset were ranked by event number". Don't you mean that it was ranked by e.g. rainfall volume and then given an event number?

Response 3.12
Thanks for pointing this out. We agree.
We will revise this and make sure to explain the rationale behind change point analysis more clearly with mathematics formulations.

Comment 3.13
Sec4.1. The removal of outliers needs to be better discussed. How do define an outlier? The theory behind seems vague.

Response 3.13
The outliers were manually selected in the old version of the manuscript, which we also realized was arbitrary.
As explained in our reply to Reviewer #1 and Review #2, we will implement the robust regression to automatically identify outliers. Particularly, robust regression works by assigning a weight to each data point. Weighting is done automatically and iteratively using a process called *iteratively reweighted least squares*. In the first iteration, each point is assigned equal weight and model coefficients are estimated using ordinary least squares. At subsequent iterations, weights are

re-computed so that points farther from model predictions in the previous iteration are given lower weight. Model coefficients are then recomputed using weighted least squares. The process continues until the values of the coefficient estimates converge within a specified tolerance.

Sec 4.1. How can you get flood reports on days with no rainfall (Zeros)? Explain the registration. 20 reports during a day with no rainfall seems strange. Are you sure that the radar worked?

Response 3.14
Reports on a day with no rainfall is mainly due to rainfall of the preceding days. We will clarify this point by giving details of the reports in the new manuscript.

Comment 3.15
Sec4.2. You study imperviousness in the same cell. What about the effect of imperviousness upstream the inundated area? Discuss.

Response 3.15:
We implemented an overland flowpath analysis, in which the effect of upstream-to-downstream flows were investigated. We will make sure to explain it more clearly.

Comment 3.16
Sec4.2. Impatient callers... How are the registration done? Are all calls registered? What are the reason for someone to call? Do they get compensation for damaged properties? Describe the registration process and discuss the implications for your analyses.

Response 3.16
We agree.
We will remove the discussion about impatient callers which does not fit here properly. Meanwhile we will add description of call registration in Section of methods.

Comment 3.17
Fig6. Did you use fine or coarse grid for these analyses?

Response 3.17
We used data per pixel (1 km x 1 km) for the analyses.

Comment 3.18
Sec5. Discuss limitations of the study. And as mentioned before, related it to others work from similar cities in the same climatic zone.

Response 3.18
As mentioned in our reply 3.1. We will expand the discussion and add comparison to similar studies.

Comment 3.19
Sec5. Spekkers 2015: You write that you found similar results before. Mention the differences between the two studies. Why did you get 7-8 mm/h in that study and 12.5 mm/h in this study? Mention the differences in data and methods used.

Response 3.19
We will add it in Discussion.
The study by Spekkers et al was based on insurance claims, instead of citizen reports. As they concluded, for rainfall events that exceed the 7–8 mm/h, failure processes in the public domain start to contribute substantially to the overall occurrence probability.

Comment 3.20
Fig1a. No need for all the details on the map, show a few city names instead. However, this is not crucial to change.

Response 3.20:
We agree.
Fig 1a will be replaced by a figure with cities names.

Comment 3.21
Fig1b&c. Scale not indicated.

Response 3.21
A scale will be added in the new version of the figure.

Comment 3.22
Fig1c. Hard to read the names.

Response 3.22
Fig 1c will be enlarged with larger fonts.

Comment 3.23
Fig2a & Fig3a. Hard to differ the marks in black and white print. For colour blind, the figure must be difficult to read at all.

Response 3.23
We agree.
To present the marks clearer, we will increase their sizes.

Comment 3.24
Fig2b & Fig3b. Difficult to see the two outliers on the left hand side.

Response 3.24
We will adjust the range of x-axis.

Comment 3.25
Spell out acronyms the first time (e.g. Dutch KNMI, German DWD and Belgium KMI).

Response 3.25
Thanks for this remark.
In the new version of the manuscript, the above-mentioned organizations will be described in full names as '*three European national meteorological services, names: the Royal Netherlands Meteorological Institute (KNMI) of the Netherlands, the German Weather Service (DWD) of Germany, and the Royal Meteorological Institute (KMI) of Belgium*'.

Comment 3.26
Sec5. paragraph 3. ", as higher rainfall amounts are more likely to cause damage to buildings." Compared to what?

Response 3.26
We agree that the statement is vague. In the new version of the manuscript, we will clarify that the rainfall depth higher than the proposed threshold may lead to a pluvial flood with higher likelihood.

References

Gaitan S, ten Veldhuis M claire, van de Giesen N (2015) Spatial Distribution of Flood Incidents Along Urban Overland Flow-Paths. Water Resour Manag 29:3387–3399. doi: 10.1007/s11269-015-1006-y

Spekkers M, Rözer V, Thieken A, et al (2017) A comparative survey of the impacts of extreme rainfall in two international case studies. Nat Hazards Earth Syst Sci Discuss 1–38. doi: 10.5194/nhess-2017-125

---

## Author Comment (AC2) · 10 Apr 2018

**Reply to interactive comments on: Towards identification of critical rainfall thresholds for urban pluvial flooding prediction based on citizen flood observations, hess-2017-751**

**Reviewer #2**

Anonymous Referee #2

Comment 2.1
Setting and boundary conditions of the study are very case specific (flat catchment, specific drainage infrastructure and operational regime (OP), rare data availability (OP data, flood reports)). Hence, transfer of findings to other cases - without having carried out similar analyses for other systems - would be, at least questionable. This clearly lowers the scientific significance of the work (reproducibility). Despite the fact that various interesting aspects of general relevance are discussed (use of citizen-reported flood incidents as ground truth data for urban flooding, change point analysis), identified (non-)correlations as well as rainfall thresholds – which relate to the key research questions in the paper - are exclusively valid for the particular case Rotterdam. I am wondering why this clearly limiting aspect has not been discussed in the manuscript. I strongly encourage the authors to address this aspect adequately, e.g. by clearly labeling results and findings as case specific (title, abstract, conclusions) and discussing the relevance of findings for other systems.

Response 2.1
Thank you very much for reviewing the paper and making helpful comments.

One of the aims of our study was to investigate the potential of citizen science data to identify flood-prone areas and thresholds that trigger flood occurrence. In addition, the thresholds can be used in a flood-early-warning context. We agree that the threshold values that we derived are specific to this case study. However, conclusions with respect to the use of this type of datasets and applicability of the threshold method are generic.

We will clarify this distinction more explicitly in the introduction, discussion and conclusions.

Comment 2.2
The title prominently suggests research on a currently popular topic: the collection and evaluation of crowdsourced data to extract meaningful information. The main text then reveals that 'crowdsourced data' are here understood as structured recordings on flood incidents reported from the public (!?) which the researchers "obtained" from an existing database (!?) – cf. page 3, line 39 -41. It is not entirely clear to me if the term "crowdsourced" refers to the fact that different people, i.e. the crowd reported the incidents or that the recordings are received from various different sources but are then formalized and archived. In a way this issue is somewhat peculiar since in previous publications the same data set had been named "citizen-reported flood incidents"... Irrespective of the fact if data used in this study can be referred to as "crowdsourced data" the novelty aspect (in the current version of the manuscript) is marginal. Hence I suggest reconsidering the title formulation or a thorough revision of the paper shifting the focus to the use of crowdsourced flood reports as such, e.g. discussing the quality of this source of information.

Response 2.2
We agreed. A similar point was raised by Reviewer #1.

We will replace the term "crowdsourced data" by the more general term "citizen observatories" and refer to (Buytaert et al. 2014; Paul et al. 2018), who summarized definitions of various citizen science-associated terms.

Comment 2.3
Using rainfall threshold values (which do not account for the spatial structure, moving patterns of storms) to warn for area-specific flooding incidents is a bit far-fetched. From my point of view, the presented results are not convincing enough to allow lumping changing factors (such as downstream drainage system behavior, operational regime, spatial rainfall variability) which eventually influence the degree of flooding and the location at which flooding occurs into a single rainfall threshold value that predicts flooding in a particular urban subcatchment. I suggest using a physically-based dual drainage model to systematically partition the influence of these aspects and to so put conclusions on a more solid basis.

Response 2.3
We agree that the spatial resolution used in our analysis to derive rainfall thresholds introduces uncertainty, as can be seen from the explanatory power that can be achieved. A higher resolution analysis for threshold identification was not possible due to limitations in the number of available data (see also results of spatial analysis in section 4.2). With the growing availability of citizen science data, more detailed analyses will be possible in the future. We will mention this point more explicitly in the discussion section.

Using a physically-based, high resolution model is unlikely to provide additional insights, because, as ten Veldhuis et al. (2010) have shown, models account for only a small part of flood-generating mechanisms in urban drainage systems. Additionally, the reliability of even very detailed physically-based models depends on availability of observational data to verify parameter assumptions. Thus far, no models have been shown to have satisfactory performance for the complex drainage system in Rotterdam.

An important motivation for our study was precisely to investigate the relationships that can be derived based on a data-driven analysis. Our investigation relies directly on citizen science data and reveals relationships between flooding and rainfall depth, an approach that is more directly accessible in many cases than complex models.

Comment: 2.4
The spatial correlation analysis between interval-specific rainfall depths and reported surface flooding observations leaves me a bit puzzled. Beside the fact that identified correlations are at the very low end in terms of occurrence and significance, the following points are at least debatable: i) As this particular part of the analysis is based on only six events I am asking how representativeness is ensured. I do not fully understand why the number of reports must be greater than 40 to allow an event to be included in the analysis. No justification is given on what the selection of this threshold is based on, nor a sensitivity analysis is conducted to show how results alter in case more events (with less reports)

are selected. In any case it should be questioned to what extend six events provide enough input for a spatial correlation analysis to come to a meaningful conclusion. ii) Disentangling dependencies: the weak but still existing correlation between population and report density suggests an inherent dependence between the two variables: the more people live in an area, the more reports can be submitted. Hence it should be discussed to what these variables can be independent at all! Depending on the way how flood reports are submitted, normalizing over population density may be a first step to research this aspect. Technically different, but similar with regard to the dependency aspect, the statement ". . . imperviousness was confirmed to be an appropriate parameter to predict urban pluvial flooding" (p. 17, line 2-3) is somewhat trivial, i.e. misleading since i) yes, sealed surfaces produce higher surface runoff and ii) solely considered, the degree of imperviousness does allow a prediction of flooding potential – it must be considered in context with other factors. A multivariate analysis approach is recommended. iii) Considering drainage network capacity constraints: the spatial analysis somehow ignores the fact that urban flooding can substantially be influenced by hydraulic behavior of the actual drainage network (e.g. hydraulic capacity constraints further downstream in the network may lead to manhole overflows). In other words: here flooding is expected to occur right where the rain cell is present, suggesting that the main cause for surface flooding is the pure amount of rainfall at the spot maybe combined with a limited capacity of street inlets. It remains an open issue to what extend the found spatial correlation is influenced/biased through this aspect (still, it is outlined in the outlook for further research – p. 17, line 37) and if this could be a reason indeed for the decreasing correlation when increasing the spatial resolution of data (finer grid). Moreover, this aspect is very likely to become more relevant when researching systems with higher terrain elevation variability, i.e. elevation difference in the catchment.

Response 2.4
We agree with your comments.

In the new manuscript, we will implement the following analyses as you suggested: a multivariate analysis, In other words, we will analyze the relationship between the number of reports and the imperviousness, rainfall intensity, and the population density. Furthermore, an analysis of variance (ANOVA) will be implemented afterwards to find the dominant factors without the need to implement normalization.

In regard to the number of reports, it is a compromise made to ensure there is at least one report per pixel. A selection of less than 40 reports leads to an event with no reports at some given pixels.

In fact, our study area is flat and the sewer (sub)districts that we consider are relatively small-sized, most of which are in a scale of five squared kilometers with multiple outflow points around the district borders. Therefore, we regard the rainfall as the main cause in the spatial analysis. We will provide more proof for this statement in Section 4.2.

Comment 2.5
The paper's layout is corrupt at many points, it seems that the manuscript had been submitted in

a rush.

Response 2.5
We realized after submission that there was problem in converting doc file into pdf file. We will pay close attention to this issue for submission next time.

Comment 2.6
Scatter plots, especially Fig. 2, 3, 5, are difficult to read. In particular outliers are difficult to spot.

Response 2.6
We agree with the fact that there is a problem of how we deal with and plot the outliers in an arbitrary way.
In the new version of the manuscript, we will implement the robust regression to eliminate the effect of outliers. Particularly, robust regression works by assigning a weight to each data point. Weighting is done automatically and iteratively using a process called *iteratively reweighted least squares*. In the first iteration, each point is assigned equal weight and model coefficients are estimated using ordinary least squares. At subsequent iterations, weights are re-computed so that points farther from model predictions in the previous iteration are given lower weight. Model coefficients are then recomputed using weighted least squares. The process continues until the values of the coefficient estimates converge within a specified tolerance.
Finally, we will compare the results of robust regression with those of linear regression (Section 4.1).

Comment 2.7
The discussion of particular events in the text referring to Fig. 4, 5 (p. 10, line 10, 22) is useless unless it is indicated in the graph and has a particular meaning.

Response 2.7
Thanks for this comment.
We will make an explanation in a better way. The effect of outliers on the regression performance will be included in the new analysis of the robust regression as described in Response 2.6.

Comment 2.8
The treatment of outliers in the OP data is occasionally fuzzy (p. 9, line 5ff; p. 10, line 10-11) and sometimes arbitrary (p. 15, line 4-6).

Response 2.8
Agreed.
We will replace this by a robust regression analysis (see the previous Response 2.6) and meanwhile we will compare the robust regression with the conventional linear regression in the paper.

Comment 2.9
The authors should revise the analysis to avoid the impression that leaving out particular data

was done to let results look a bit better.

Response 2.9
Agreed.
Applying a robust regression to the dataset will ensure objective identification of outliers.

References

Buytaert W, Zulkafli Z, Grainger S, et al (2014) Citizen science in hydrology and water resources: opportunities for knowledge generation, ecosystem service management, and sustainable development. Front Earth Sci 2:1–21. doi: 10.3389/feart.2014.00026

Paul JD, Buytaert W, Allen S, et al (2018) Citizen science for hydrological risk reduction and resilience building. Wiley Interdiscip Rev Water 5:e1262. doi: 10.1002/wat2.1262

---

## Author Comment (AC3) · 10 Apr 2018

Reply to interactive comments on: Towards identification of critical rainfall thresholds for urban pluvial flooding prediction based on citizen flood observations, hess-2017-751

Reviewer #1

***Comment 1.1:*** *Does the novelty/contribution lie in the use of crowdsourced data? If so, the paper is not written as if it is. If the novelty is indeed in the use of crowdsourced data, then the paper should focus on the crowdsourced data and make more of a discussion/examination of the use of the data to make a strong case of its associated difficulties/advantages. As it is, the introduction and lit review are only in a general sense with no focus on crowd sourcing. And only about 1/3 of the methods and results are based on crowdsourced data*

**Response 1.1:**

The main novelty of our work does not lie in the use of crowdsourced data, but rather in the formulation of a data-driven approach to predict urban pluvial flooding without the need to run hydrodynamic models. An important outcome of this approach is the derivation of critical rainfall thresholds, which can be used to predict flood occurrence and test the performance of urban stormwater systems. To clarify the novelty of our approach, we will highlight this more explicitly in Introduction (subsection 1.3).

***Comment 1.2:*** *I am not sure if the historical complaint reports used can be called "crowdsourced". The reports are in the order of 10-100 per day (as inferred by Fig. 2). However, in today's context, crowdsourcing commonly refers to sourcing from a large pool of people using the Internet, smartphones, surveillance cameras etc. to obtain observations in the hundreds to thousands to millions. Thus, it may be misleading to describe the observations referred to in this paper as crowdsourced.*

**Response 1.2:**

Thanks for this comment. In fact, many definitions of crowdsourcing and related terms like citizen science and citizen observatories can be found in the literature, covering a wide range of data sources and collection methods. After reviewing the literature we agree that the term citizen observatories better fits the nature of our dataset than crowdsourcing, following definitions given in (Buytaert et al., 2014; Herman Assumpção et al., 2017). Hence, we will adopt this terminology instead.

***Comment 1.3:*** *There could be a temporal resolution mismatch between the flood complaint reports used and a storm event. That is, in a storm event, it may not be possible to determine*

*which reports coincide with the peak rainfall. Thus, there are some uncertainties in the "crowdsourced" observations that are nontrivial and may affect the validity of the methods/results.*

**Response 1.3:**

We conducted our analyses at daily resolution, as citizen reports were captured at this temporal resolution; sub-daily resolution information was not available. Within the daily time window, we considered the maximum rainfall depth at three different temporal resolutions: 15 minutes, 60 minutes and 24 hours. This approach is similar to the one adopted, for instance, by (Spekkers et al., 2015). Given that flood response in highly impervious areas like the districts in Rotterdam is very short (less than an hour up to maximum a few hours), the gap between rainfall peak and flood observation is likely to be small. We will emphasize this particularity of urban flood response better in the Methods section (subsection 3.2).

*Comment 1.4: The paper found a strong correlation between surface imperviousness and the number of flooding reports and concluded that "there is some explanatory power behind degree of imperviousness as an urban pluvial flooding parameter". However, the correlation is likely a spurious correlation due to the correlation between population density and the number of flooding reports, and population density and imperviousness. Thus, this conclusion of the paper is not well-justified. The may be advisable for the authors to use other methods (e.g. multivariate linear regression) to exclude the effects of population density and arrive at a more justifiable conclusion.*

**Response 1.4:**

Thanks for this valuable comment, we agree that multivariate analysis will add value to the study. In the revised version, we will add results from a multivariate regression analysis. In other words, we will analyze the relationship between the number of reports and the imperviousness, rainfall intensity, and the population density. Furthermore, an analysis of variance (ANOVA) will be added to find the dominant factors explaining variability in the number of flood observations.

*Comment 1.5:   Other minor points: 1. Page 4, Line 7, $km^2$ instead of km2 should be used. 2. Page 5, Line 10, I guess the area of the green roof park should be 40,000 m2 instead of 40.000 m2. 3. Page 6, Line 25, equation (1) should be appeared before equation (2).*

**Response 1.5:**

Thanks for pointing out this issue, we will correct this.

*Comment 1.6:  The authors should also remove duplicated flooding reports in their temporal and spatial correlation analysis.*

**Response 1.6:**

In the Results section, we explained our method to identify and delete duplicated flooding reports (subsection 4.2, page 13). However, this explanation should have been placed in the Methods Section; this will be corrected in the new version of the manuscript.

*Comment 1.7:  Instead of using the rainfall intensity at the center of the study area for the temporal correlation analysis, I would suggest the authors to use the aerial average rainfall intensity of the whole study area. This is to take into account the rainfall spatial variability, which could be rather high at the scale of the study area.*

**Response 1.7:**

We agree with this comment, which was also raised by Reviewers #2 and #3.

We will redo the analysis, using an aerial average rainfall intensity, instead of rainfall over the central radar pixel.

*Comment 1.8: What is the unit of dt, mm or mm/h? It seems to be mm/h according to the definition in equation (1), but is shown to be mm in Figures 2, 3, and 4. The authors should also provide more clarified explanations for the variables in equation (1).*

**Response 1.8:**

The coefficient $d_t$ stands for the rainfall depth within a given time window and has unit mm. The time windows $t$ that we adopted are 15 minutes, 60 minutes and 24 hours, respectively. So they are expressed as $d_{0.25}$, $d_1$ and $d_{24}$ in terms of hours.

We will make this clearer in the revised version.

References

Buytaert, W., Zulkafli, Z., Grainger, S., Acosta, L., Alemie, T.C., Bastiaensen, J., De Bièvre, B., Bhusal, J., Clark, J., Dewulf, A., Foggin, M., Hannah, D.M., Hergarten, C., Isaeva, A., Karpouzoglou, T., Pandeya, B., Paudel, D., Sharma, K., Steenhuis, T., Tilahun, S., Van Hecken, G., Zhumanova, M., 2014. Citizen science in hydrology and water resources: opportunities for knowledge generation, ecosystem service management, and sustainable development. Front. Earth Sci. 2, 1–21. doi:10.3389/feart.2014.00026

Herman Assump ção, T., Popescu, I., Jonoski, A., Solomatine, D.P., 2017. Citizen observations contributing to flood modelling: opportunities and challenges. Hydrol. Earth Syst. Sci. Discuss. 1–26. doi:10.5194/hess-2017-456

Spekkers, M.H., Clemens, F.H.L.R., ten Veldhuis, J.A.E., 2015. On the occurrence of rainstorm damage based on home insurance and weather data. Nat. Hazards Earth Syst. Sci. 15, 261–272. doi:10.5194/nhess-15-261-2015